# Policy shaping based on the learned preferences of others accounts for risky decision-making under social observation

**HeeYoung Seon, Dongil Chung***

Department of Biomedical Engineering, Ulsan National Institute of Science and Technology, Ulsan, Republic of Korea

## eLife Assessment

Seon and Chung investigate changes in own risk-taking behavior, when they are being observed by a "risky" or "safe" player. Using computational modeling and model-informed fMRI, the authors present **convincing** evidence that participants adjust their choice congruent with the other player's type (either risky or safe). The conclusions of the paper are an **important** contribution to the field of social decision-making as they show a differentiated adjustment of choices and not just a universally riskier choice behavior when being observed as has been claimed in previous studies.

***For correspondence:**
dchung@unist.ac.kr

**Competing interest:** The authors declare that no competing interests exist.

**Abstract** Observing others' choices influences individuals' decisions, often leading them to follow others. However, it is repeatedly reported that being observed by others tends to make people act more riskily. We hypothesized that this discrepancy arises from individuals' belief that others prefer riskier choices than they do. To examine this hypothesis, we used a gambling task where on some trials, individuals were informed that their choices would be observed by a risk-averse or seeking partner. Most important, individuals were given chances to learn each partner's preference beforehand. As expected, individuals initially believed that partners would make relatively riskier choices than they would. Against two alternative explanations, we found that individuals simulated partners' choices and weighed these simulated choices in making their own choices. Using functional magnetic resonance imaging, we showed that decision probabilities adjusted with the simulated partners' choices were represented in the temporoparietal junction (TPJ). Moreover, individual differences in the functional connectivity between the TPJ and the dorsomedial prefrontal cortex (dmPFC) were explained by the interaction between model-estimated social reliance and sensitivity to social cues in the dmPFC. These findings provide a neuromechanistic account of how being observed by others affects individuals' decision-making, highlighting the roles of the dmPFC and TPJ in simulating social contexts based on individuals' beliefs.

## Introduction

It is well-known that individuals often make choices differently around social others (*Abrams and Hogg, 1990*; *Cikara et al., 2011*; *Guassi Moreira et al., 2018*). Although the most commonly observed social influence is to conform with the choices observed from social others (*Hardin and Higgins, 1996*), another line of research has shown that the mere presence of social others biases individuals' choices to become riskier (*Gardner and Steinberg, 2005*; *Haddad et al., 2014*). This unidirectional behavioral bias was particularly observed in adolescents (*Chein et al., 2011*; *Ciranka and van den Bos, 2019*), and thus previous studies largely focused on the characteristics observed in adolescents (e.g. heightened social sensitivity, *Albert and Steinberg, 2011*; *Foulkes and Blakemore,*

**eLife digest** People, especially adolescents, often act differently around their peers than around their parents, and tend to take more risks when they are with peers. People usually copy the behavior of others and make similar choices. But previous research suggests that humans are also likely to take more risks when others observe them. Scientists think that making decisions could be influenced by what people think they know or believe about other people's preferences.

Seon and Chung tested the hypothesis that people are influenced by both the choices of peers and by their beliefs about the preferences of those around them. To examine this, they designed an experiment in which participants made decisions while being observed by one of two different partners. Before the experiment began, participants had the chance to observe each partner's decisions and thereby learn that one tended to take risks while the other preferred to play it safe. By analyzing both their choices and their brain scan data, they investigated whether people adjusted their decisions depending on the observing partners' preferences.

Before learning about the preferences of others, participants thought that anonymous observers would choose riskier options than they themselves would. Through repeated observation, however, they discovered which partner tended to take more risks and which tended to play it safe. This led participants to adjust their own choices depending on which partner was watching.

Computational modeling showed that people were not simply copying the observer's behavior. Instead, they mentally simulated what the partner would have done in the same situation, based on what they had learned, and let these predictions guide their decisions. Brain scans revealed that two brain regions involved in social cognition – and the strength of the connection between these regions – explained individual differences in how much people relied on these predictions when making choices.

These regions, the temporoparietal junction and dorsomedial prefrontal cortex, are often referred to as the social brain and are thought to play a key role in the mental simulation of other people's intentions. People who relied more on their own predictions about observers showed greater sensitivity in these regions. The connection between the regions increased or decreased depending on whether a partner was observing.

This study suggests that our choices are shaped by what we think an observer would do. These insights help explain everyday behaviors, such as why people post differently on social media when they expect certain reactions, or why opinions change in group settings. Such knowledge can shed light on social pressure and online influence. Still, because real-life situations are much more complex, further research in natural social environments is needed.

---

*2016*; *Lundborg, 2006*) and developmental imbalance between reward sensitive system and cognitive control system (*Chein et al., 2011*; *Somerville et al., 2010*) to explain why being under this type of social context (presence of social others) affects decision-makers in a seemingly different manner. However, recent studies showed that the unidirectional influence of social others' presence can also be observed in adults (*Otterbring, 2021*), and that the extent of this influence depends on the observer's identity—specifically, whether the observer is a parent or a peer (*van Hoorn et al., 2018*). These data suggest that besides the neurodevelopmental characteristics, there exists an active processing of information about the social context which determines how individuals respond to others' presence. Expanding this perspective, it can be inferred that the beliefs individuals have about social others, which can be changed and established by learning, may have a crucial role in determining the direction of social influence. Yet, this hypothesis about the impacts of social others' presence has not been explicitly tested.

To examine how individuals' beliefs about social observers affect their decisions about risky options, we used a three-phased gambling task (*Figure 1a*, *Figure 1—figure supplement 1*) in which 43 healthy participants (male/female = 25/18, age = 21.35 ± 2.42; *Supplementary file 1A*) made choices between a safe option (i.e. guaranteed payoff) and a risky option. In the first phase of the task ('Solo phase'), individuals were asked to make a series of gambling choices alone (*Figure 1e*). In the second phase ('Learning phase'), individuals were asked to predict gamble choices of the two random partners who were introduced as previously participated players whose choices were recorded (*Figure 1b and*

*f*). This phase was expected to provide individuals to learn about the two partners where unbeknownst to participants, one partner was set to be risk-averse and the other partner was set to be risk-seeking (see *Figure 1—figure supplement 2* for the partners' preferences). In the third phase ('Observed phase'), individuals were given the same set of gamble choices they faced in the Solo phase, each pair of gambles iterated three times in total, but all shuffled in a random order (*Figure 1g*). Critically, on some trials, individuals were told that their choices would be used for one of the partners' Learning phase. In this way, we implemented two types of trials where the choices would be observed by the partners (Risk-averse and seeking observer trials) and one type of trials where the choices would be made alone (No observer trials). By examining individuals' choice patterns in the Learning phase, we aimed to examine the impacts of observers on individuals' risky decision-making.

Previous functional neuroimaging studies on decision-making under social contexts revealed a set of brain regions that have critical roles in processing social information (*Blakemore, 2008*; *Hiser and Koenigs, 2018*; *Mukerji et al., 2019*). For example, the dorsomedial prefrontal cortex (dmPFC) is known to encode social information not specific to valuation (*Amodio and Frith, 2006*; *Mitchell et al., 2006*; *van Overwalle, 2009*), but also known to encode value signals estimated in perspectives of others (*Behrens et al., 2008*; *Ruff and Fehr, 2014*; *Sul et al., 2015*; *Wittmann et al., 2016*). Such neural instantiation of social valuation is dissociable from the patterns observed in the ventromedial

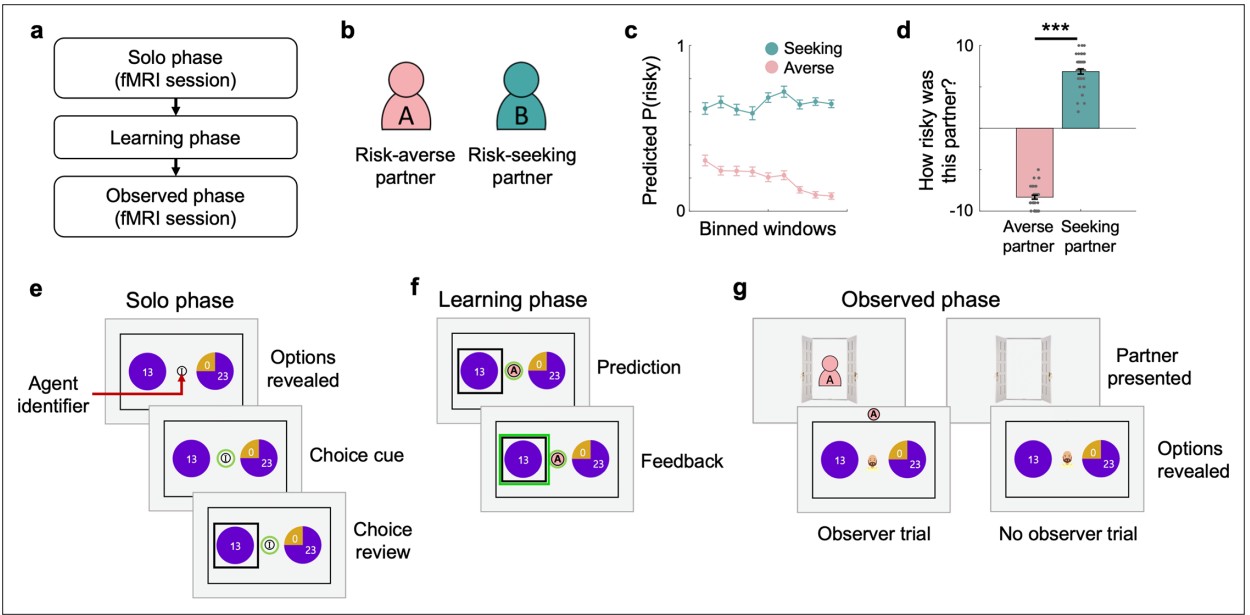

**Figure 1.** Experimental paradigm. (**a**) The task comprised three phases: Solo, Learning, and Observed phases. (**e**) During the Solo phase, participants were asked to make a series of risky choices alone, which were used to measure their own risk preference. (**a**, **b**, **f**) During the Learning phase, participants were introduced to two random partners and asked to predict their choices. Unbeknownst to participants, one partner had risk-aversive (Risk-averse partner) and the other partner had risk-tolerant (Risk-seeking partner) preferences. To help partner identification, each partner was labeled with an alphabet letter (A or B) and color-coded (counterbalanced). On each trial, an agent identifier that indicates the identity of the predicted partner was presented on the center of the screen. (**a**, **g**) During the Observed phase, participants were asked to make the same type of gamble choices as the Solo phase. Critically, at the beginning of some trials ('Observer trial'), participants were informed that their choice on the corresponding trial would be later used in the Learning phase for one of the two assigned partners. On these Observer trials, the identity of the designated partner on each trial was presented as an avatar observing through an open door. 'No observer trials', the trials at which individuals' choices will not be presented to any partners, were informed with a vacant open door. (**c**) To depict individuals' prediction performance during the Learning phase, participants' prediction choices were binned with a bin size of six trials and three-trial overlaps. Along the repeated trials of prediction with feedback, individuals successfully learned the two partners' simulated risk preferences. Error bars indicate s.e.m. (**d**) At the end of the Learning phase, individuals were asked to answer a few questions regarding their impression about each partner's characteristics. Participants' reports on the question 'How risky was this partner?' showed a consistent pattern with their prediction behavior, such that they evaluated the Risk-seeking partner to be significantly riskier than the Risk-averse partner ($t(42)=-35.83$, $p=4.10e-33$). Gray dots represent each individual's evaluation score; Error bars indicate s.e.m.; ***$p<0.001$.

The online version of this article includes the following figure supplement(s) for figure 1:

**Figure supplement 1.** Task timeline for each task phase.

**Figure supplement 2.** Estimated model parameters in the Solo phase.

prefrontal cortex (vmPFC), which is a region known to track both the subjective valuations of non-social monetary rewards and the combined values of their own preferences with their preferences for (or aversion to) social contexts (*Amodio and Frith, 2006*; *Hare et al., 2010*; *Mitchell et al., 2006*; *van Overwalle, 2009*). The temporoparietal junction (TPJ) is another region known to play an important role in a range of social cognitive functions, including simulating others' intention and choices, as well as learning about others (*Behrens et al., 2008*; *Boorman et al., 2013*; *Charpentier et al., 2020*; *Park et al., 2021*; *Samson et al., 2004*; *Saxe and Kanwisher, 2003*; *Saxe and Kanwisher, 2013*; *van Overwalle, 2009*; *Young et al., 2010*). Corroborating these neuroimaging findings, excitatory stimulation of the TPJ improved social cognition involving the control of self-other representation (*Santiesteban et al., 2012*), while TPJ dysfunctions in psychiatric patients (e.g. schizophrenia and autism spectrum disorder) were associated with their social impairments (*Carter and Barch, 2007*; *Lombardo et al., 2011*). Previously, it was shown that when others' choices were revealed, this network of brain regions is involved in inferring others' intentions and using the social information in the process of decision-making (*Hampton et al., 2008*; *Zhang and Gläscher, 2020*). We hypothesized that, even in the absence of explicit information about others' choices, the mere presence of social others could lead participants to conform to the option they believe others would choose. To do so, participants would need to simulate others' potential choices, particularly when option values vary across trials. As a result, we propose that the same brain regions involved in simulating others' decisions would also be engaged during value-based decision-making in the presence of social observers.

To test our hypotheses, we scanned a subset of 30 participants (male/female = 16/14, age = 21.77 ± 2.16; *Supplementary file 1A*) and used a computational modeling approach in conjunction with neuroimaging to investigate the mechanisms via which the presence of social observer affects individuals' decision processes. Behaviorally, we first tested whether individuals successfully learned partners' choice preferences and then investigated whether their decisions, made under the belief of being observed by others, could be accounted for by a combination of their own preferences and the simulated choice tendencies of the observers. Neurally, we predicted that the choices made under social observation would be signaled both in the valuation-association regions (vmPFC, dmPFC) and the region associated with social inference (TPJ). Our results provide neural and behavioral evidence for a role of social observers in affecting individuals' decisions.

## Results

### Individuals initially believe that others would make riskier choices than they would

As the main purpose of the current study, we aimed to investigate the impacts of two different observing partners, each of whom has different risk preferences, on individuals' risky decision-making. To examine individuals' own risk preferences before they were exposed to any potential social influence, all participants were asked to make a series of gamble choices by themselves in the initial phase of the task (Solo phase; *Figure 1a and e*). After the measurement of individuals' preferences, they were introduced to two random partners and asked to predict these partners' gambling choices (Learning phase; *Figure 1b and f*). Since individuals were not provided with any information about either of the partners, we assumed that individuals' very first predictions about partners' choices might reflect their initial beliefs about others' preferences. Compared with the choices that individuals were expected to make based on their Solo phase choice patterns (i.e. simulated choices based on individuals' risk preferences), they reported that partners might make riskier choices (Wilcoxon signed-rank test = 66, p=0.0495; *Figure 2a*). This result indicates that individuals may initially believe anonymous others to be more risk-seeking than themselves.

Across repeated attempts to predict partners' choices with feedback, it was confirmed that individuals successfully learned to dissociate and predict the choices each partner would have made (see Materials and methods for details; *Figure 1c*). Furthermore, individuals' prediction responses during the subsequent 10 trials without feedback consistently indicated that they could distinguish between the two partners and accurately estimate each partner's risk preferences (t(42) = –11.46, p=1.66e-14; *Figure 2b*). Self-reported ratings of the partners' perceived riskiness, collected after the Learning phase, further supported this finding (t(42) = –35.83, p=4.10e-33; *Figure 1d*). These results suggest

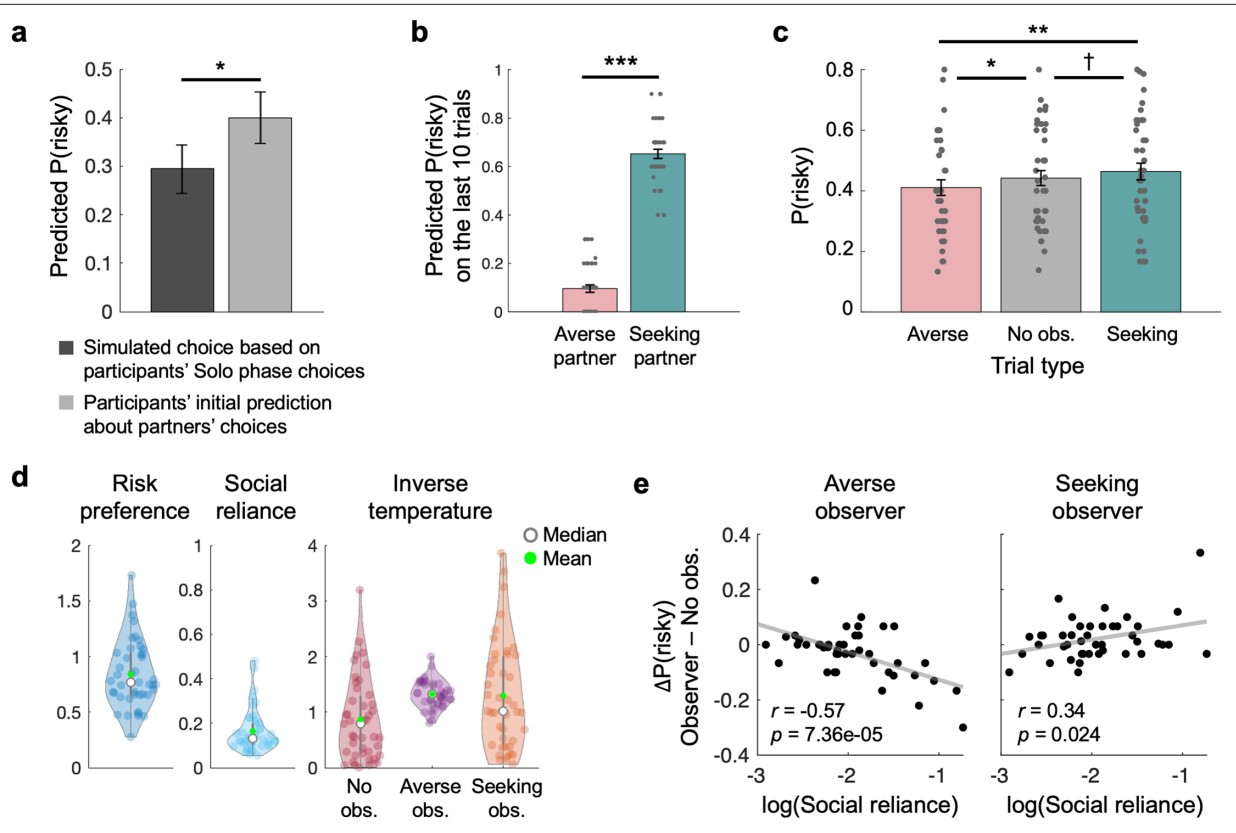

**Figure 2.** Behavioral results. (**a**) During the Learning phase, individuals were asked to predict partners' choices. On the very first prediction trial, individuals had to make predictions without any information about partners. Compared to the choices that individuals would have made based on their estimated risk preferences, individuals predicted that the partners were more likely to choose the risky option (Wilcoxon signed-rank test W=66, p=0.0495). (**b**) Individuals did not receive feedback about their predictions on the last 10 trials in the Learning phase. On these trials, participants still predicted that one partner whose true risk preference is set to be riskier than the other partner would make riskier choices, and vice versa (t(42) = –21.54, p=2.56e-24). Each dot represents an individual participant. (**c**) In the Observed phase, individuals made gambling choices under three different conditions: Risk-averse observer, No observer, and Risk-seeking observer trials. Relative to No observer trials, participants made more safe gambles when the risk-averse partner was to observe their choices, and more risky gambles when the risk-seeking partner was to observe (repeated-measures ANOVA, F(2, 42)=7.26, p=0.0012; paired t-tests: Risk-averse vs. No observer: t(42) = –2.28, p=0.028; Risk-seeking vs. No observer: t(42) = 1.84, p=0.072; Risk-averse vs. Risk-seeking: t(42) = –3.28, p=0.0021). Error bars indicate s.e.m.; †p<0.1, *p<0.05, **p<0.01. ***p<0.001. (**d**) Model parameters were estimated using the Social reliance model. Light-colored dots represent individuals. Filled green dots and empty markers indicate means and medians of each parameter, respectively. (**e**) Estimated Social reliance parameters well explained individuals' choices during the Observed phase. Specifically, individuals who relied the most on the observers' choice tendencies chose the risky option the least when the Risk-averse partner would be observing, but chose the risky option the most when the Risk-seeking partner would be observing. Each dot represents an individual participant, and solid lines indicate regression lines.

The online version of this article includes the following figure supplement(s) for figure 2:

**Figure supplement 1.** Formal model comparison and identifiability evaluation.

**Figure supplement 2.** Relationship between risk preferences in the Solo and the Observed phases.

**Figure supplement 3.** Individuals' impressions of the partners and their impact on subsequent choices.

that independent of the initial beliefs about social others, individuals can learn about others' preferences through experience.

## Social observation from safe and risky partners affects individuals' choices

During the Observed phase, individuals were instructed that some of their choices would be 'observed' (see Materials and methods for the instruction) by one of the two partners whose risk preferences they had learned about in the Learning phase ('Observer trial'; *Figure 1g*). Compared with the trials where the choices were made alone without any observers ('No observer trial'), individuals were

indeed affected by learned preferences of the observing partners (repeated-measures ANOVA, $F_{(2, 42)}=7.26$, p=0.0012; *Figure 2c*). Specifically, under the observation of the risk-averse partner, individuals chose the risky gamble significantly less than they did under no social observation. Under the observation of the risk-seeking partner, individuals were swayed toward making riskier choices, such that the probability of choosing the risky over the safe gamble was marginally larger than that under no observation and significantly larger than that under the risk-averse partner's observation (Risk-seeking vs No observer: $t(42) = 1.84$, p=0.072; Risk-seeking vs Risk-averse: $t(42) = 3.28$, p=0.0021; Risk-averse vs No observer: $t(42) = -2.28$, p=0.028; *Figure 2c*). Participants' average probability of choosing the risky gamble under no social observation was comparable to that measured during the Solo phase ($t(42) = 0.68$, p=0.50), which suggests that the observed changes in choices are indeed the results of social observation rather than task repetition. These results indicate that social observation affects individuals' decisions, and that the direction of this social influence depends on individuals' beliefs about others.

## Individuals' simulated choices of social observers shape how they make risky choices

Previous studies have shown that social context can influence decision-making processes. However, the underlying mechanisms proposed have varied depending on how the social information was presented. For example, when the choices of social others were explicitly revealed, observing the choices added additional utilities on the options chosen by others and swayed individuals to follow others (*Chung et al., 2015*; *Chung et al., 2020*). On the contrary, when individuals were given the chance to learn about one other player's risk preference, without being explicitly shown her/his choices, their own preferences tended to shift closer to the learned preferences of the other player, ultimately leading to more similar choices (*Suzuki et al., 2016*). Here, similar to the latter case, participants did not see what others chose. However, the current study differed from both previous cases in that individuals were informed that their choices would be observed by others whose preferences they had previously learned.

We hypothesized that if individuals are sensitive to the learned preferences of the observing partner, they would use this information to simulate the partner's likely choices, rather than simply aligning their own preferences with those of the partner. To test our hypothesis, we constructed a computational model assuming that individuals mix their choice tendencies based on their own preferences and the simulated choice tendencies derived from the perceived (or belief about) preferences of the observing partner. Specifically, we estimated a 'Social reliance' parameter ($\omega$) that accounts for the extent to which the simulated choice tendencies of others ($\text{Prob}_{\text{Partner}}$; *Equation 2*) contribute to individuals' decisions ($\text{Prob(risky)} = (1 - \omega) \text{Prob}_{\text{Own}} + \omega \text{Prob}_{\text{Partner}}$; *Equation 3*; see Materials and methods for details). Note that the tendencies of how individuals ($\text{Prob}_{\text{Own}}$; *Equation 1*) or partners ($\text{Prob}_{\text{Partner}}$; *Equation 2*) make risky choices were formalized using a power utility function (each option's utility $U = \sum p_i (v_i)^{\rho}$ per expected utility theory (*Kahneman and Tversky, 1979*) for a gamble that has an $i^{\text{th}}$ outcome $v_i$ with probability $p_i$ for an individual whose risk preference is $\rho$) and the softmax decision rule (*Huettel et al., 2006*; *Preuschoff et al., 2006*). In one extreme where $\omega=0$, the model converges to $\text{Prob}_{\text{Own}}$, indicating that individuals do not change how they make risky choices even under the social context (i.e. being under others' observation). The other extreme ($\omega=1$) makes the model equivalent to $\text{Prob}_{\text{Partner}}$, indicating that individuals always choose according to the risk preference they believed the partner holds.

A formal model comparison against two alternative models—the Risk preference change model (*Suzuki et al., 2016*) and the Other-conferred utility model (*Chung et al., 2015*); see Materials and methods for model descriptions—confirmed that individuals' choice behaviors were best explained by our suggested Social reliance model (*Figure 2—figure supplement 1a*). We simulated individual choice data for 43 simulated subjects assuming the usage of each of the three potential models, and by re-estimating model parameters, we confirmed that the simulated behaviors from each model were best explained by the corresponding model (i.e. model recovery; *Figure 2—figure supplement 1b*). In addition, all parameters were recoverable (risk preference [ $\rho$ ]: r=0.98, p<0.001; social reliance [ $\omega$ ]: r=0.43, p=0.0038; value sensitivity for No observer trials [ $\lambda_{\text{NoObs}}$ ]: r=0.86, p<0.001; value sensitivity for Risk-averse observer trials [ $\lambda_{\text{AverseObs}}$ ]: r=0.49, p<0.001; value sensitivity for Risk-seeking observer trials [ $\lambda_{\text{SeekingObs}}$ ]: r=0.72, p<0.001; see Materials and methods for model and parameter recovery details).

The estimated Social reliance parameter ($\omega$) showed a consistent pattern with model-agnostic measures that capture the extent to which individuals were influenced under partners' observation (Averse observer: r = –0.57, p=7.36e-05, Seeking observer: r=0.34, p=0.024; *Figure 2d and e*). Specifically, individuals who relied the most on partners' simulated choice tendency (i.e. largest $\omega$) showed the largest shift toward making safer choices under the risk-averse partner's observation compared to the choices they made under no observation (r=–0.57, p=7.36e-05), and showed the largest shift toward riskier choices under the risk-seeking partners' observation (r=0.34, p=0.024). Note that the rank order of individuals' risk preference parameters ($\rho$) was highly consistent between contexts with and without social observation (r=0.73, p=2.14e-08; *Figure 1—figure supplement 2*, *Figure 2—figure supplement 2a*). However, individual differences in risk preference were more pronounced under partners' observation ($\chi^2$=18.90, p=1.45e-05; *Figure 2—figure supplement 2b*), such that the risk preference of individuals who made riskier (or safer) choices alone shifted toward stronger risk-seeking (risk-aversion) under social observation (r=0.33, p=0.031; *Figure 2—figure supplement 2c*). These results suggest that there are two independent impacts of social observation on risky decision-making; individuals' own risk preferences are more pronounced, independent of the identity of the observing partner, and furthermore, individuals' choices are shaped by their beliefs about the currently observing partner's preferences. Corroborating these model-based results, individuals' self-reports about the impression they had on partners (e.g., similarity, trustworthiness), collected after the Learning phase (*Supplementary file 1I*), were consistent with these parameterized impacts of social observation (*Figure 2—figure supplement 2*, *Figure 2—figure supplement 3*).

## Neural responses support social reliance and the effect of social observation on shaping individuals' final choices

To investigate neural instantiation of decision processes under social observation, we tested two neural hypotheses constructed based on our suggested computational model. First, if individuals followed our model, we would expect to find neural representations that reflect individuals' final choices. To set regions-of-interest (ROIs) for analyses in the Observed phase, we first analyzed individuals' blood-oxygen-level dependent (BOLD) responses at the time at which they viewed choice options during the Solo phase and sought the brain regions that track trial-by-trial decision probabilities (i.e. model-estimated Prob(chosen); see DM0 in Materials and methods). We found that the ventromedial prefrontal cortex (vmPFC; x = –3, y=62, z = –13, $k_E$ = 165, cluster-level $P_{FWE, SVC}$ = 0.009), ventral striatum (vStr; x=3, y=14, z = –10, $k_E$ = 40, cluster-level $P_{FWE, SVC}$ = 0.015), and dorsal anterior cingulate cortex (dACC; x=12, y=32, z=29, $k_E$ = 386, cluster-level $P_{FWE, SVC}$ = 0.005)—brain regions known to be involved in valuation and decision-making (*Boorman et al., 2013*; *Christopoulos et al., 2009*; *Croxson et al., 2009*; *Kable and Glimcher, 2007*; *Knutson and Bossaerts, 2007*; *Rangel and Hare, 2010*; *Rudebeck et al., 2008*; *Rushworth et al., 2011*; *Wunderlich et al., 2009*)—tracked final decision probabilities in the Solo phase (*Figure 3a*; *Supplementary file 1B*). These results were largely the same when the trial-by-trial utility differences were used as a parametric modulator instead (*Figure 3—figure supplement 1*; *Supplementary file 1C*). From the subsequent ROI analyses in the Observed phase, consistent with the results in the Solo phase, BOLD responses from the ROIs significantly tracked individuals' final decision probabilities (vmPFC: $t$(29) = 1.77, p=0.044, vStr: $t$(29)=3.21, p=0.0016, dACC: $t$(29) = –4.07, p=0.00017; *Figure 3b*; see ROI analyses in Materials and methods). See *Figure 3—figure supplement 2* for the neural representations of one's own utility and that of the observer, as estimated by the Social reliance model.

Second, we expected that, under social observation, brain regions distinctive from those engaged during the Solo phase would be additionally recruited. A subsequent whole-brain analysis (see DM1 in Materials and methods) revealed additional brain regions that tracked trial-by-trial model-estimated final decision probabilities during the Observed phase (*Supplementary file 1D*). Notably, bilateral TPJ showed significant positive tracking of decision probabilities (left TPJ: x = –54, y = –37, z=14, $k_E$ = 64, $P_{unc.}$ <0.001; right TPJ: x=63, y = –40, z=17, $k_E$ = 191, cluster-level $P_{FWE, SVC}$ = 0.019; *Figure 3c*). The involvement of the TPJ in social contexts is consistent with previous studies that showed its role in social cognitive functions, including mentally simulating others' motives (*Behrens et al., 2008*; *Boorman et al., 2013*; *Charpentier et al., 2020*; *Park et al., 2021*). No additional regions survived whole-brain correction.

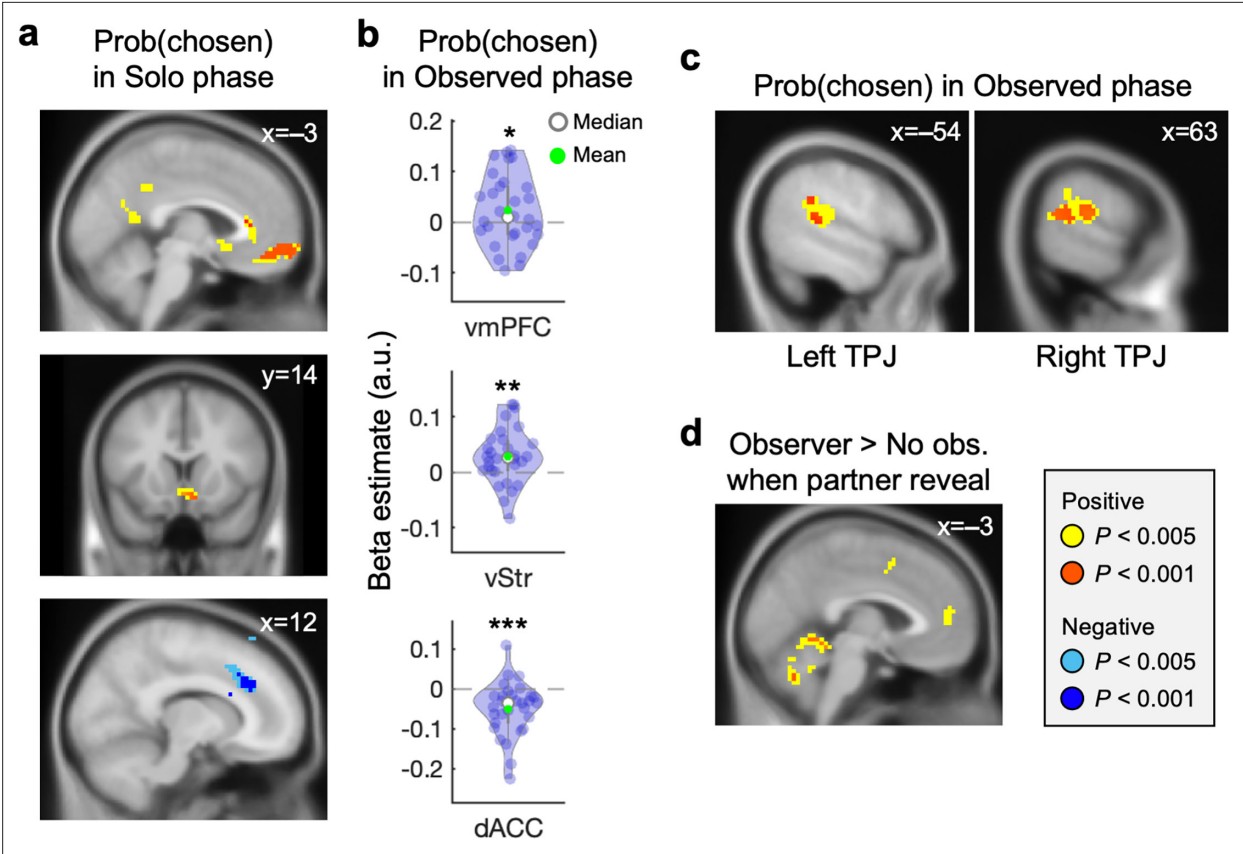

**Figure 3.** dmPFC and TPJ are recruited for valuation under social observation in addition to the regions tracking non-social subjective value. (**a**) When viewing gamble options during the Solo phase, trial-by-trial probability of the chosen option was positively encoded in the vmPFC (x = –3, y=62, z = –13, $k_E$ = 165, cluster-level $P_{FWE,\ SVC}$ = 0.009) and vStr (x=3, y=14, z = –10, $k_E$ = 40, cluster-level $P_{FWE,\ SVC}$ = 0.015), and negatively encoded in the dACC (x=12, y=32, z=29, $k_E$ = 386, cluster-level $P_{FWE,\ SVC}$ = 0.005). These brain regions were set as regions-of-interest (ROI) for the decision-making signals in the Observed phase where gambling choices were identical besides the social context. (**b**) To examine whether the same decision-tracking regions were recruited in the Observed phase, trial-by-trial probability of the chosen option was calculated based on our suggested Social reliance model. As expected, the same type of decision probability information comprising the social and non-social components was tracked in the ROIs during the Observed phase. Each dot represents an individual participant.; *p<0.05; **p<0.01; ***p<0.001. (**c**) Whole brain analysis revealed that trial-by-trial probability of the chosen option was positively encoded in the bilateral TPJ when individuals were viewing gamble options during the Observed phase (left TPJ: x = –54, y = –37, z=14, $k_E$ = 104, $P_{unc.}$ <0.001; right TPJ: x=63, y = –40, z=17, $k_E$ = 191, cluster-level $P_{FWE,\ SVC}$ = 0.019). (**d**) An additional whole-brain analysis revealed that the dmPFC responded to the initial social cue (x = –3, y=50, z=14, $k_E$ = 22, $P_{unc.}$ <0.005).

The online version of this article includes the following figure supplement(s) for figure 3:

**Figure supplement 1.** Neural substrates of trial-by-trial encoding of utility differences between the chosen and unchosen options.

**Figure supplement 2.** Representation of one's own utility and that of the observer along the ventral-to-dorsal axis of the mPFC.

**Figure supplement 3.** Neurosynth meta-maps for the term "decision.".

**Figure supplement 4.** Neurosynth meta-maps for non-valuation social processing.

At the beginning of each trial in the Observed phase, individuals were first cued with the presence (or absence) of an observing partner (*Figure 1g*, *Figure 1—figure supplement 1c*). This cue was intended to dissociate neural responses to the social context per se (i.e. the presence of an observing partner), which we hypothesized would initiate social processing, from the neural processes involved in incorporating this information during the subsequent decision-making phase. A subsection of the dmPFC showed higher responses on Observer compared to No observer trials, indicating that the region is involved in social processing when necessary ('dmPFC_contrast' hereafter; x = –3, y=50, z=14, $k_E$ = 22, $P_{unc.}$ <0.005; *Figure 3d*; *Supplementary file 1E*; see DM3 in Materials and methods). We tested whether the dmPFC was also involved in incorporating social information during the decision process under social observation, particularly among individuals who relied more heavily on simulating others' behavior. To examine this hypothesis, we conducted a psychophysiological interaction

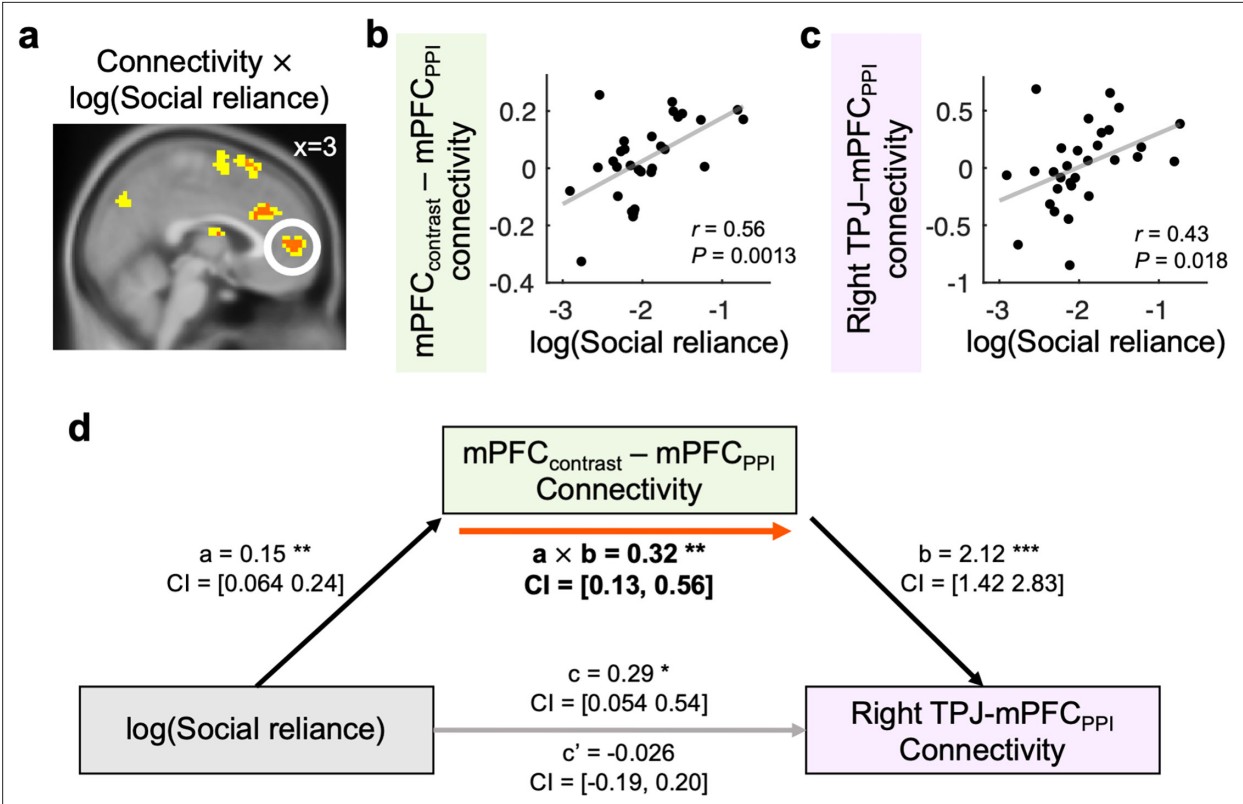

**Figure 4.** TPJ-dmPFC connectivity is associated with individuals' social reliance. To examine whether the dmPFC$_{PPI}$ and the TPJ interacted with each other while individuals made choices under social observation, we conducted psychophysiological interaction (PPI) analyses. (**a**, **b**) The functional connectivity between the dmPFC$_{contrast}$ from **Figure 3d** and its adjacent, anatomically distinct region within the dmPFC region (dmPFC$_{PPI}$) was positively associated with log-transformed Social reliance (peak at [x=3, y=50, z=5], $k_E$ = 74, cluster-level $P_{FWE, SVC}$ = 0.011). The clusters displayed in yellow $P_{unc.}$ <0.005 and red $P_{unc.}$ <0.001. (**c**) Additional PPI analysis between the TPJ from **Figure 3c** and the dmPFC$_{PPI}$ from **a** (a region sensitive to the initial social cue) was also positively associated with log-transformed Social reliance (r=0.43, p=0.018). (**d**) The positive relationship between individuals' social reliance and TPJ-dmPFC$_{PPI}$ connectivity was mediated by the dmPFC$_{contrast}$-dmPFC$_{PPI}$ connectivity (a: $\beta$=0.15, p=0.0013, b: $\beta$=2.12, p=1.42e-06, a×b: $\beta$=0.32, p=0.0016). Black and gray arrows indicate significant and non-significant associations between the components, respectively. Red arrow indicates a significant mediation effect; *p<0.05; **p<0.01; ***p<0.001.

The online version of this article includes the following figure supplement(s) for figure 4:

**Figure supplement 1.** Left TPJ-dmPFC$_{PPI}$ connectivity is associated with individuals' social reliance.

(PPI) analysis where the dmPFC$_{contrast}$ identified above (**Figure 3d**; **Supplementary file 1E**) was set as a seed, [Observer trials – No observer trials] was set as a psychological factor, and the model-estimated Social reliance parameter was entered as a covariate (see Materials and methods for PPI design). We confirmed that the functional connectivity between the dmPFC$_{contrast}$, which is sensitive to cues regarding the presence of an observing partner, and its adjacent, anatomically distinct region within the dmPFC ('dmPFC$_{PPI}$' hereafter; x=3, y=50, z=5, $k_E$ = 74, cluster-level $P_{FWE, SVC}$ = 0.011; **Figure 4a and b**; **Supplementary file 1F**) was positively associated with individuals' social reliance.

Per our Social reliance model, the simulated behavior of others is integrated into the final decision probability. We found that the final decision probability was represented in the TPJ specifically under social observation. If the dmPFC$_{PPI}$ region from the above PPI result (**Figure 4a**; **Supplementary file 1F**) plays a role in simulating the other's behavior, it should interact with the TPJ to integrate the social information into the final decision. To examine this hypothesis, we conducted another PPI analysis where the TPJ was set as a seed region and the target region was the dmPFC$_{PPI}$ cluster identified in **Figure 4a** (see Materials and methods for details). Indeed, the TPJ-dmPFC$_{PPI}$ connectivity was positively associated with individuals' social reliance (r=0.43, p=0.018; **Figure 4c**, **Figure 4—figure supplement 1a**). That is, particularly for individuals who relied strongly on other observers' preferences, TPJ-dmPFC$_{PPI}$ connectivity was significantly stronger under others' observation than when

making choices alone. Consistent with previous findings on the TPJ (*Behrens et al., 2008*; *Boorman et al., 2013*; *Charpentier et al., 2020*; *Park et al., 2021*; *Young et al., 2010*), our model-based analyses indicate that the TPJ plays a key role in inferring others' behavior and using that information to make individuals' own decisions.

Supporting the role of the TPJ-dmPFC$_{PPI}$ connectivity in social information processing, the association between individuals' social reliance and the connectivity was mediated by the extent to which individuals responded to the initial social cue (dmPFC$_{contrast}$-dmPFC$_{PPI}$ connectivity, mediation effect: $\beta$=0.32, p=0.0016; see Materials and methods for the mediation analysis; *Figure 4d*, *Figure 4—figure supplement 1b*). This links our model-based and neuroimaging results. Our data suggest that individuals who were predisposed to rely on belief about others' choices were more sensitive to the social cue, and that individuals who exhibited higher dmPFC sensitivity to social cue showed stronger TPJ-dmPFC connectivity in processing social information. Together, our results suggest that the dmPFC has a critical role in relaying both social and non-social valuation information for final decision-making processes. Given that our sample size is smaller than the recommended threshold for detecting mediation effects (*Fritz and MacKinnon, 2007*), this significant indirect effect should be interpreted with caution, particularly with respect to causal inference.

## Discussion

Being around social others is known to affect how individuals behave (*Abrams and Hogg, 1990*; *Cikara et al., 2011*; *Guassi Moreira et al., 2018*). Here, we developed a novel experiment where individuals made risky choices under the observation of social others after learning about their risk preferences. Individuals did not always act riskier under social observation, but rather tended to adjust their choices toward the direction they believed the observing partner would prefer. By using a computational model-based approach, we showed that the choice tendencies simulated based on the belief about the observer's preference contribute to individuals' decision-making processes. Our results provide a mechanistic explanation for how the presence of social observers influences individuals' decision-making processes.

Observational learning and mimicry of others' behavior are patterns commonly found in social animals, including nonhuman primates (*van de Waal et al., 2013*). Such behaviors are thought to be driven either by a motivation to acquire additional information ('informational conformity') or by a motivation to align with group norm ('normative conformity'), even when doing so does not necessarily lead to better outcomes (e.g. higher accuracy; *Cialdini and Goldstein, 2004*). Given that there are no objectively correct or incorrect answers in the gambling task used in our study, the observed social influence is more consistent with normative conformity. However, we cannot rule out the possibility that individuals developed false beliefs about a particular observing partner—namely, that the partner had greater control over or insight into the gambling task. Future studies are needed to directly investigate whether individuals' beliefs about others modulate informational social influence—that is, their motivation to use social information to gain additional insight by inferring others' potential choices.

Regardless of the underlying motivation for social influence, its direction is generally expected to be bidirectional, such that individuals may become either more or less averse to an option they originally preferred (*Suzuki et al., 2016*). In contrast to this possibility of bidirectional social influence, the presence of social others has been shown to exert a unidirectional effect on individuals' choices, such that particularly among adolescents, who tend to make riskier choices when they are in the presence or under the observation of peers (*Powers et al., 2018*). The current study suggests that this seemingly contradicting finding may be accounted for by the beliefs individuals hold about social others. Before learning about others, individuals may perceive themselves as decision-makers who make stable and safe choices, and thus, might attempt to make riskier choices when others are watching. However, after explicitly learning about others' preferences, their initially *misplaced* beliefs about others would be adjusted, which can then be incorporated into decision-making processes under social observation. Our computational framework provides an explanation for why and how social influence on individuals' choices sometimes manifests as a unidirectional shift of their preferences, while at other times, it results in a bidirectional shift. Future studies may directly examine the beliefs of adolescents about their peers and test whether such biased beliefs (e.g. 'All the cool kids would not be afraid of jumping off the cliff.') indeed drive their riskier behaviors around peers (*Gardner and Steinberg, 2005*; *Haddad et al., 2014*).

By using a computational modeling approach, we were able to delineate the impact of social influence across various cognitive processes embedded during decision-making (e.g. valuation, action selection). Previously, it was suggested that an observation of explicitly presented others' choices adds value to the option others chose during the process of valuation (*Chung et al., 2015*). Under social others' observation, no explicit choice information was provided, and thus we expected that the same mechanism would not explain the influence of social observation. Our data suggest that, as expected, being under social observation does not alter individuals' valuation of the choice options, but instead alters their decision-making policies to mimic the behavior of others. This result may seem contradictory to *Suzuki et al., 2016*, a previous study that examined the impact of another type of implicit social information. In the corresponding study, authors suggested that repeated opportunities to predict others' choices have an effect of swaying individuals' risk preferences toward the learned preferences of others, a mechanism that we rejected for the current study. We note that such a mechanistic discrepancy is likely due to the differences in task design between the research and ours. Specifically, our study provided a comparable setting for learning about others (the Learning phase herein), but then implemented an additional layer of social influence (i.e. social observation), which was the main social factor we investigated. Our data suggest that individuals who have previously acquired knowledge about the social observer tend to rely on simulated choices which would be made by the observer, leading to a fine shaping of their decision policy.

Our modeling results were corroborated with the vmPFC response that tracked individuals' final decision probabilities under social observation. Various decision-making studies suggested that the vmPFC encodes subjective value signal (*Clithero and Rangel, 2014*; *Levy and Glimcher, 2012*), which also encompasses the value of social information (*Chung et al., 2015*) and the expected value across temporal horizon (*Iigaya et al., 2020*; *Na et al., 2021*). Our result expands this view and suggests that the vmPFC tracks individuals' decision policies (i.e., final decision probabilities), specifically when their choices are adjusted for the extent to which they rely on what social others would do. Typically, individuals' decisions are tightly linked to their subjective valuation, making it hardly possible to dissociate the representation pattern of one from the other. In the current study, as shown in our Social reliance model, the decision-making process under others' observation clearly drifts apart from the form of combining information in value level. By separately testing these two possibilities, our neural results provide evidence for an alternative view, suggesting that the vmPFC sometimes takes part in tracking action policies rather than tracking economic values per se (*Hayden and Niv, 2021*). Note that this does not rule out the broadly accepted role of the vmPFC as the subjective value encoder, but instead opens the possibility that the brain region may have a more generalizable capacity. Depending on the context, individuals may make (subjective) value-based choices or follow alternative heuristics (e.g. relying on belief about others' choices). Independent of such specific cognitive processes, the vmPFC may combine information from multiple sources and track individuals' behavioral intentions.

Inference about social others is known to recruit a set of brain regions including TPJ and dmPFC (*Amodio and Frith, 2006*; *Behrens et al., 2008*; *Boorman et al., 2013*; *Charpentier et al., 2020*; *Mitchell et al., 2006*; *Park et al., 2021*; *Ruff and Fehr, 2014*; *Samson et al., 2004*; *Saxe and Kanwisher, 2003*; *Saxe and Kanwisher, 2013*; *Sul et al., 2015*; *van Overwalle, 2009*; *Wittmann et al., 2016*; *Young et al., 2010*). Among the so-called 'Social brain' regions, previous studies showed that the TPJ is known to take part in theory-of-mind (TOM), the ability to understand others' intention and to attribute others' mental states to oneself (*Saxe and Kanwisher, 2013*; *Schurz et al., 2017*). Consistent with the known functional roles of the brain region, particularly when individuals believed others were observing their choices, the TPJ tracked individuals' decision policies, indicating that the region plays a key role in simulating others' choices and using that information to make their own decisions. Furthermore, in line with the known roles of the dmPFC in the valuation process when making choices for others (*Sul et al., 2015*) and in processing information obtained from social others (*Chung et al., 2020*), our results show that the functional communication between the TPJ and dmPFC is crucial for this social simulation.

The current study provides a mechanistic account for how the presence of others and others' observing influence individuals' risky decision-making. Our Social reliance model provides a way in which individuals adjust their choices around others and may explain why adolescents exhibit well-behaved behavior with their parents but are more likely to act out when with their peers (*van Hoorn et al., 2018*). When others' choices are not explicitly provided but rather presented as a context

that individuals must infer, their tendency to rely on social information contributes to changes in their actions than changes in valuation. In the modern world, as much as we have means to obtain information about others, almost every choice we make is seen by others. Under this inevitable social environment, taking perspectives of others is largely adaptive and essential ability to empathize with others' pain (*de Vignemont and Singer, 2006*; *Decety and Lamm, 2006*; *Lamm et al., 2007*), make prosocial choices (*Buckholtz et al., 2008*; *Nook et al., 2016*), and follow social norm (*Cialdini and Goldstein, 2004*; *Rilling and Sanfey, 2011*). Our data shed light on the flip side, elucidating why and how incorrect beliefs about others and excessive or biased sensitivity to social context may lead to maladaptive behaviors, such as a higher tendency to commit crimes (*Buck et al., 1989*) and the formation of extremely polarized opinions (*Geschke et al., 2019*; *Lefebvre et al., 2024*).

## Materials and methods

### Participants

53 individuals participated in the current study, 38 of whom were scanned while they were making decisions. All participants provided written informed consent. This study was approved by the Institutional Review Boards of the Ulsan National Institute of Science and Technology (UNISTIRB-19–45 A). One participant was excluded due to excessive movement (>3 mm in the x, y, or z direction or >2 degree rotation around x, y, or z axis) and two participants were excluded due to scanner artifact. Seven participants (five from the scanned and two from the behavior-only participants) were additionally excluded due to their failure to learn and distinguish the two partners' risk preferences in their post-task questionnaires (the self-reported risky level difference between the two partners was smaller than [median – 3*(median absolute deviation)], i.e., judging the risk-averse partner as more risky than the risk-seeking partner). After exclusion, data from 43 participants (male/female = 25/18, age = 21.35 ± 2.42) were used for behavioral analyses, and a subsample of the data (N=30; male/female = 16/14, age = 21.77 ± 2.16; *Supplementary file 1A*) were used for neuroimaging analyses.

### Experimental procedures

We developed a novel three-phase gambling task (*Figure 1*, *Figure 1—figure supplement 1*). Throughout the task, participants were asked to make a series of choices between one certain and one risky gamble option. Each option was presented as a pie chart. The certain option was presented as a whole pie, indicating a 100% probability of earning the payoff written on top of the pie. The risky option was divided into two pie pieces, with the size of each piece indicating the probability of earning high- and low-payoffs written on top of each piece. First, 30 sets of gambles were generated (*Supplementary file 1G*). The gamble set was created with nine unique risky options, combining high-payoffs of 25, 48, and 90 with probabilities of 0.25, 0.5, and 0.75, as in a previous study (*Chung et al., 2017*). The low payoff was always zero. Certain payoffs were set below, close to, and above the expected value of each risky option. In addition, one additional set of gambles was added with the largest expected value difference (EV difference = 38.5). Two sets of gambles were added as 'catch trials' (to ensure participants' attention) where the payoff of the certain option was larger than the high-payoff of the risky option. Second, 30 sets of gambles were additionally generated in the same manner (*Supplementary file 1H*). Nine unique risky options were created by pairing three levels of high-payoffs (15, 34, and 90) and three probabilities of high-payoff (0.25, 0.5, and 0.75). Certain payoffs were set as described above, but they were tailored so that the distribution of the expected value differences between the certain and risky options matched the first 30 sets of gambles. The first 30 sets of gambles were used for the first phase ('Solo') and the third phase ('Observed') of the task, the second 30 sets of gambles were used for the second phase ('Learning').

In the Solo phase, participants completed 30 gamble choices. These choices were used to measure individuals' own risk preferences without the influence of social context. In the Learning phase, participants were randomly matched with two partners and asked to predict their choices. The partners were introduced as individuals who previously participated in the gambling task. Unbeknown to the participants, one of the two partners was set to be risk-averse (risk preference = 0.43) and the other to be risk-seeking (risk preference = 1.12; see Partners' choices and *Figure 1—figure supplement 2* for details). There were 40 prediction trials (20 trials for each partner) that provided feedback, followed by 20 additional prediction trials (10 trials for each partner) without feedback. The order of the partners

to be predicted on each trial was randomized. On the last 20 prediction trials, individuals were asked to report their confidence level on a 10-point Likert scale (1: not confident at all, 10: very confident) after making their choices. At the end of the Learning phase, participants answered eight questions about their impressions of each predicted partner. The questions included likability, trustworthiness, riskiness, attractiveness, competence (metacognition, general academic grades), consistency, and preference similarity (see *Supplementary file 1I* for the full list of questions). One additional question was included as an attention check ('Answer 5'). The order of these questions was randomized.

In the Observed phase, participants played another gambling task. Most importantly, individuals were informed that their gamble choices on some trials would be observed by the partners they were matched with in the Learning phase ('Observer trial'). Specifically, we instructed participants that the partners would revisit the lab and predict the corresponding gamble choices of the participants during their Learning phase, and that the partners would answer the eight impression questions about the participants (*Supplementary file 1I*). Informing which partner would observe the choice, at the start of every Observer trial, the identity of a partner was presented in the center of the screen (*Figure 1g*). On the other trials, participants were informed that their choices would not be observed by anyone ('No observer trial'), which was indicated with an empty open door presented at the beginning of the trials. No observer and Observer trials were intermixed, and the Observer trials comprised trials with the risk-averse partner and the risk-seeking partner. In total, each participant had 90 trials (30 unique gamble sets × 3 trial types [No observer; Risk-averse observer; Risk-seeking observer]).

At the end of the study, participants were paid the base rate (20,000 Korean won per hour) and the bonus (2300–7010 Korean won), determined by the outcome of a random single gamble drawn from the Solo phase, the outcome of another gamble from the Observed phase, and the performance during the Learning phase.

## Partners' choices

Two partners' choices over the 30 sets of the Learning phase gambles were generated and provided as feedback during the phase. We first analyzed behavioral choices of 74 non-overlapping participants on the Solo phase gamble sets. Assuming that individuals' gamble choices are accounted for by expected utility theory (*Bernoulli, 1954*), we used a standard power utility function (Utility $U = \sum P_i(V_i)^\rho$) and softmax choice rule (Probability(Risky) = $[1 + \exp(-\mu (U_{Risky} - U_{Certain}))]^{-1}$), and estimated their risk preferences ($\rho$) and value sensitivities ($\mu$). To simulate one partner who is risk-averse and one partner who is risk-seeking, we selected the risk-preference parameters that are [mean ± 1.5 × standard deviation of the risk preference distribution]: risk-averse $\rho$ =0.43, risk-seeking $\rho$ =1.12 (*Figure 1—figure supplement 2*). The two partners' choice behaviors on the gamble sets in the Learning phase were simulated by using a standard power utility function, value sensitivity $\mu$=5, and each partner's risk preference.

## Behavioral analyses

To compare individuals' own choice tendencies with their initial belief about partners, we simulated the choices that participants would have made for the same set of gambles, and used a Wilcoxon signed-rank test to compare these choices against the initial predictions (assumed as a fixed effect across partners). To simulate individuals' choices, we used their estimated risk preferences from the Solo phase. To track the changes in individuals' beliefs about the partners' risk preferences, we binned their predictions about each partner (window size = 6 trials, overlap = 3 trials) and calculated the proportion of risky choices for each bin (*Figure 1c*). In addition, the proportion of risky choices in individuals' predictions during the last 10 trials in the Learning phase was calculated for each partner and compared to examine whether they successfully learned and distinguished the risk preferences of the two partners. For model-agnostic behavioral analyses, we used repeated measures ANOVA and compared the proportion of risky choices among Observer trials with the risk-averse partner, No observer trials, and Observer trials with the risk-seeking partner. Paired t-tests were used for post-hoc analyses between each pair of trial types. After conducting model-based analyses (see Computational modeling below), we examined their consistency with model-agnostic measure by calculating Pearson's correlation coefficient between individuals' model-based Social reliance parameters and the impact of observation on the proportion of risky choices (i.e. the differences between Observer and No observer trials). *The MathWorks Inc, 2019* (MathWorks) was used for all statistical tests of

behavior. All statistical tests were two-tailed with an alpha level of 0.05 unless noted otherwise. The dataset and analysis scripts generated in this study are available at OSF (https://doi.org/10.17605/OSF.IO/3VBDA; *Chung and Seon, 2025*).

## Computational modeling

For model-based analyses, choices in the Solo phase were used to estimate individuals' risk preferences, and choices in the Observed phase were used to examine the mechanisms by which individuals were affected by social observation. Individuals' predictions in the Learning phase were used to estimate their learned beliefs about each partner.

## Solo phase

As noted above and per expected utility theory (*Bernoulli, 1954*), we used a standard power utility function (Utility $U = \sum P_i(V_i)^\rho = P_{\text{high-payoff}} \times V^\rho$) and softmax choice rule (Prob(Risky) = $[1 + \exp(-\mu(U_{\text{Risky}} - U_{\text{Certain}}))]^{-1}$) to estimate individuals' risk preference $\rho$ and value sensitivity μ. Note that $0 < \rho < 1$ captures risk-aversive choices, $\rho = 1$ captures risk-neutrality, and $\rho > 1$ captures risk-seeking choices.

## Learning phase

Individuals could use the feedback provided during the first 20 trials in the Learning phase to learn about each partner's risk preference. We used individuals' prediction choices for each partner during the last 10 trials in the Learning phase to estimate their beliefs about the partners' risk preferences. As in the Solo phase analysis, we used a standard power utility function and softmax choice rule to explain individuals' risky choices. Estimation of the beliefs about the partners' risk preferences ($\rho_{\text{partner, risk-averse}}$, $\rho_{\text{partner, risk-seeking}}$) was conducted with custom MATLAB script using maximum log-likelihood estimation (MLE) at the individual subject level (maximum of 25,000 function evaluation and iterations). The custom analysis scripts generated in this study are available at OSF (https://doi.org/10.17605/OSF.IO/3VBDA; *Chung and Seon, 2025*).

## Observed phase

We hypothesized that under social observation, individuals may (i) implement the belief about the partners' preferences in making choices, (ii) change their preferences to match that of the partners, or (iii) alter their valuation depending on the identity of the currently observing partner. For a formal model comparison, we computed the integrated Bayesian Information Criteria (iBIC) to examine whether either of the alternative models explains individuals' behavior equally well or better than the Social reliance model (a smaller iBIC score indicates a better model fit) (*Figure 2—figure supplement 1a*).

### Social reliance model

Our hypothesis was that individuals simulate the choices of the social observer and take into account this simulated choice tendency when making their own choices. Individuals had the opportunity to learn about the partners' preferences (Learning phase) and might use these beliefs to simulate the choices that each partner would make. Constructed based on the same power utility function and softmax choice rule, individuals' choice tendencies were defined in two components:

$$\text{Prob}_{\text{own}}(\text{Risky}) = \left[1 + \exp\left(-\mu_{\text{own}}\left(U_{\text{Risky, own}} - U_{\text{Certain, own}}\right)\right)\right]^{-1} \tag{1}$$

$$\text{Prob}_{\text{partner}}(\text{Risky}) = \left[1 + \exp\left(-\mu_{\text{partner}}\left(U_{\text{Risky, partner}} - U_{\text{Certain, partner}}\right)\right)\right]^{-1} \tag{2}$$

where $U_{\text{own}}$ and $U_{\text{partner}}$ are subjective values (utilities) calculated with individuals' own risk preference $\rho_{\text{own}}$ and the learned risk preference of the currently observing partner ($\rho_{\text{partner, risk-averse}}$ or $\rho_{\text{partner, risk-seeking}}$), respectively. $\mu_{\text{own}}$ is individuals' value sensitivity between the two options, and depending on the identity of the observing partner, $\mu_{\text{partner}}$ is separated into $\mu_{\text{risk-averse}}$ and $\mu_{\text{risk-seeking}}$. Individuals' final choices under social observation were defined to be determined based on a weighted mixture between the two decision probabilities:

$$\text{Prob}(\text{Risky}) = (1 - \omega) \times \text{Prob}_{\text{own}}(\text{Risky}) + \omega \times \text{Prob}_{\text{partner}}(\text{Risky}) \tag{3}$$

where $\omega$ is the Social reliance parameter (defined between 0 and 1) that represents the extent to which individuals rely on others' choice tendencies. The Social reliance model included five free parameters: $\rho_{own}$, $\mu_{own}$, $\mu_{risk-averse}$, $\mu_{risk-seeking}$, $\omega$. Note that belief about the partners' risk preferences was estimated from individuals' predictions in the Learning phase.

## Social risk preference change model

If individuals alter their risk preferences under social observation as reported in a previous study (*Suzuki et al., 2016*), their choices in the Observed phase would be captured by separate risk preferences, each of which assigned to a specific type of trial: $\rho_{own}$ for No observer trials, $\rho_{partner, risk-averse}$ for Observer trials with the risk-averse partner, and $\rho_{partner, risk-seeking}$ for Observer trials with the risk-seeking partner. Including a common value sensitivity µ for all choice trials, the Social risk preference change model included 4 free parameters: $\rho_{own}$, $\rho_{partner, risk-averse}$, $\rho_{partner, risk-seeking}$, µ.

## Other-Conferred Utility (OCU) model

Previously, an explicit presentation of social others' choices added value to the choice option that others chose (*Chung et al., 2015*; *Chung et al., 2020*). Although the partners' choices were not explicitly revealed, the social context of being under observation may also alter individuals' valuation of options. To capture this possibility, we defined two 'other-conferred utility (OCU)' terms that represent additional values added to either the certain or the risky option, depending on the observer's identity. Note that individuals successfully learned to distinguish the two partners' risk preferences, and thus might use the identity information in valuation. Specifically, we assumed that under the risk-averse partner's observation, individuals would add value to the certain option (*Equation 4*), whereas under the risk-seeking partner's observation, they would add value to the risky option (*Equation 5*).

$$\text{Prob(Risky)} = \left[1 + \exp\left(-\mu\left(U_{Risky} - \left(U_{Certain} + OCU_{risk-averse}\right)\right)\right)\right]^{-1} \tag{4}$$

$$\text{Prob(Risky)} = \left[1 + \exp\left(-\mu\left(\left(U_{Risky} + OCU_{risk-seeking}\right) - U_{Certain}\right)\right)\right]^{-1} \tag{5}$$

The OCU model included four free parameters: $\rho$, $OCU_{risk-averse}$, $OCU_{risk-seeking}$, µ.

## Parameter estimation

Parameters were estimated using a hierarchical Bayesian model estimation (*Ahn et al., 2017*; *Daw, 2011*) and Markov chain Monte Carlo (MCMC) sampling with the No-U-Turn variation of the Hamiltonian Monte Carlo technique implemented in Stan and its interface to R (*Team SD, 2020*). For all parameters, the group-level distributions were assumed to be Gaussian with free hyperparameters of group-level mean, standard deviation (SD), and a standard normal distribution (*Normal*(0, 1)) following noncentered parameterization (*Team SD, 2020*). For µ, $\rho$, and $\omega$, we applied an inverse probit transformation and then multiplied a constant (50 for all µs [µ, $\mu_{own}$, $\mu_{risk-averse}$, $\mu_{risk-seeking}$], 2 for all $\rho$s [$\rho$, $\rho_{own}$, $\rho_{partner, risk-averse}$, $\rho_{partner, risk-seeking}$], and 1 for $\omega$) to constrain the parameters between 0 and the multiplied constant. OCU parameters were not constrained. We estimated the hyperparameters of the group-level distributions using uninformative priors: means ~*Normal*(0, 10) and SDs ~*Cauchy*(0, 2.5) with lower bound of zero.

## Model and parameter recovery

To examine whether our model alternatives are identifiable in the empirical parameter range, we condu9cted a model recovery analysis (*Wilson and Collins, 2019*). First, the choices on the same set of gambles in the task were simulated for each model alternative using the parameters estimated from 43 participants' empirical data. Second, the simulated data were fitted to all model alternatives, and iBIC scores were calculated to identify the best explanatory model. Based on 10 iterations of these procedures, the proportion of the best explanatory model was calculated for each model alternative and depicted as a confusion matrix (*Figure 2—figure supplement 1b*). In addition, we conducted a parameter recovery analysis to assess whether model parameters in the best explanatory model (Social reliance model) were identifiable from each other (*Wilson and Collins, 2019*). Specifically, we simulated the gamble choices using the parameters ($\rho_{own}$, $\omega$, $\mu_{own}$, $\mu_{risk-averse}$, $\mu_{risk-seeking}$) estimated from 43 participants' empirical data (true parameter), and then re-estimated the model parameters

(recovered parameter). To examine identifiability of the parameters, we calculated Pearson's correlation between the true and recovered parameter pairs (*Figure 2—figure supplement 1c*).

## Neuroimaging acquisition and preprocessing

Functional and structural MRI brain scans were acquired with a Siemens MAGNETOM TRIO 3 T scanner. High-resolution T1 weighted structural images were acquired through magnetization prepared-rapid gradient echo (MP-RAGE) sequence with the parameters: repetition time (TR)=2300 ms, echo time (TE)=2.28 ms, slices = 192, voxel size = $1.0 \times 1.0 \times 1.0$ mm$^3$, flip angle = 8°, field of view (FOV)=256 mm. Echo planar images were collected during the Solo and Observed phases to measure blood oxygen-level-dependent (BOLD) signal. Scans were angled 30° from the anterior commissure–posterior commissure line. The functional images were acquired with the parameters: repetition time (TR)=2000 ms, echo time (TE)=20 ms, slices = 44, voxel size = $3.0 \times 3.0 \times 3.0$ mm$^3$, flip angle = 80° field of view (FOV)=192 mm. The functional images were preprocessed using MATLAB and Statistical Parametric Mapping (SPM) 12 (https://www.fil.ion.ucl.ac.uk/spm/). The preprocessing analysis included slice-timing correction, realignment, co-registration, segmentation, spatial normalization to the Montreal Neurological Institute (MNI) template, and smoothing using an 8 mm full width at half maximum (FWHM) Gaussian kernel. A high-pass filter of 1/128 Hz was applied to all sca,ns and autocorrelation of the hemodynamic responses was modeled as a first-order autoregressive process.

## General linear model (GLM) analyses

We performed event-related fMRI analyses of the BOLD responses during the Solo and Observed phases. One design matrix (DM) was used for the Solo phase to assess the brain regions that track trial-by-trial decision probabilities (see DM0 below). Four DMs were used for the Observed phase: two for assessing the impacts of social observation on individuals' decision tendencies and subjective valuation (see DM1, DM2), and two for investigating the functional interactions between brain regions dependent on social observation (see DM3, DM4). For each design matrix, realignment parameters were included to model movement artifacts.

In DM0, all choice trials in the Solo phase were modeled as linear regressors. The task-related regressors were as follows:

1. *Fixation*: a crosshair presentation that indicates the initiation of a new choice trial
2. *ViewOptions*: revelation of new pair of gambles
3. *ChoiceCue*: choice cue presentation informing individuals that a keypress response is enabled
4. *Keypress*: all gamble choices during the decision period

Neural responses to *ViewOptions* were modeled as 6 s events and the responses to other events were modeled as stick functions. The events in each regressor were convolved with the canonical hemodynamic response function and its temporal and dispersion derivatives. To assess the neural instantiation of final decision tendencies, parametric modulators associated with trial-by-trial decision probabilities for the chosen options (Prob(chosen)) were calculated using each individual's parameters estimated from the Social reliance model, and were applied to *ViewOptions*. The parametric modulator was z-score transformed.

To assess neural responses in the Observed phase, DM1 and DM2 were constructed, modeling all choice trials in the Observed phase as linear regressors. Reflecting the task-design, the task-related regressors were as follows:

1. *Fixation*: a crosshair presentation that indicates the initiation of a new choice trial
2. *ViewPartner*: revelation of the identity of the partner who will be observing the current choice trial [No observer, Risk-averse partner, or Risk-seeking partner]
3. *ViewOptions*: revelation of new pair of gambles
4. *ChoiceCue*: choice cue presentation informing individuals that a keypress response is enabled
5. *Keypress*: all gamble choices during the decision period

Neural responses to *ViewPartner* and *ViewOptions* were modeled as 3 s and 6 s events, respectively, and the other events were modeled as stick functions. In the same way as DM0, all the events were convolved with the canonical hemodynamic response function and its temporal and dispersion derivatives. In DM1, parametric modulators associated with Prob(chosen) were applied to *ViewOptions*. In DM2, to examine the neural substrates of subjective valuations from individuals' own and

their partners' perspectives, two sets of parametric modulators associated with the utility differences between chosen and unchosen options (ΔU) were applied to *ViewOptions*. One set was calculated using individuals' own risk preference $\rho_{own}$, and the other set was calculated using the learned risk preference of the observing partner ($\rho_{partner, risk-averse}$ or $\rho_{partner, risk-seeking}$). Each set of the parametric modulator was z-score transformed.

We constructed two additional design metrics to investigate the brain regions that process the context of social observation. In DM3, to investigate the neural substrates sensitive to social observation, we separated *ViewPartner* in DM1 into two separate event regressors: one for when the choices were observed by either the risk-averse or the risk-seeking partner (*ViewPartner_Observer*) and one for when the choices were not observed by any partners (*ViewPartner_NoObserver*). Other regressors and settings were kept the same as in DM1. To examine whether the functional connectivity between brain regions was associated with the social context, we constructed DM4 to have the same structure with DM1, except *ViewOptions* was divided into Observer events (*ViewOptions_Observer*) and No observer events (*ViewOptions_NoObserver*). Note that this was to use [Observer trial – No observer trial] as a psychological factor in our subsequent psychophysiological interaction analyses (see Psychophysiological interaction analyses below).

Contrast images were generated for each individual at the first level, and at the second level, one-sample t-tests were conducted to estimate the group average response or the association between individuals' BOLD responses and their model estimated Social reliance parameters. For the second-level regression, we added individuals' Social reliance parameter as a covariate to DM4. The family-wise error rate (FWE) with small-volume correction (SVC) was used for multiple comparisons (see ROI analyses).

## Region-of-interest (ROI) analyses

To define ROIs for the neural analyses conducted in the Observed phase, we used significant clusters identified during the Solo phase. Specifically, regions showing significant activation for Prob(chosen) in the DM0 (thresholded at p<0.001) were selected as ROIs. Three ROI clusters were defined: the vStr (peak voxel at [x=3, y=14, z = –10], $k_E$ = 9), vmPFC (peak voxel at [x = –3, y=62, z = –13], $k_E$ = 99), and dACC (peak voxel at [x=12, y=32, z=29], $k_E$ = 118). These ROIs were then applied in the Observed phase analyses to test whether similar neural representations are also engaged in social contexts.

## Term-based meta-analytic maps from Neurosynth for small volume correction

To reduce the likelihood of false positives arising from random significant activations and to enhance sensitivity within regions of theoretical interest, small volume correction (SVC) was applied using term-based meta-analytic maps from Neurosynth. This approach allows for hypothesis-driven correction by restricting statistical testing to anatomically and functionally defined ROI. Specifically, three meta-analytic maps were generated using Neurosynth's term-based analyses (*Yarkoni et al., 2011*), with a false discovery rate (FDR) corrected p<0.01 and a cluster size >100 voxels. For each resulting cluster, we defined a spherical ROI with a 10 mm radius centered on the cluster's center of gravity. For each anatomically distinct brain region, only a single center of gravity was identified and used to define the corresponding ROI.

First, to identify regions encoding final decision probabilities during the Solo phase and enhance sensitivity, we used the meta-map associated with the term 'decision' to identify neural substrates of value-based decision-making. This yielded three clusters: vmPFC ([x = –3, y=38, z = –10]), vStr ([x=12, y=11, z = –7]), and dACC ([x=3, y=26, z=44]) (*Figure 3a*, *Figure 3—figure supplement 3*). Second, to examine social processing during the Observed phase, we used the meta-map associated with the term 'social' to identify brain regions typically involved in social cognition. This analysis revealed clusters, including the rTPJ ([x=51, y = –52, z=14]) and lTPJ ([x = –51, y = –58, z=17]) (*Figure 3c*, *Figure 3—figure supplement 4a*). Third, to define an ROI involved in processing social cues independent of valuation, we used a meta-map associated with 'social' but excluding 'value', isolating regions specific to non-valuation-related social cognition. This analysis revealed a cluster, including the dmPFC ([x=0, y=50, z=14]) (*Figures 3d and 4a*, *Figure 3—figure supplement 4b*).

## Psychophysiological interaction (PPI) analyses

We tested whether the functional connectivity between brain regions was associated with individuals' social reliance. We conducted two psychophysiological interaction (PPI) analyses during the time at which the options were presented. First, to investigate brain regions that are sensitive to social cues (or context), we used the dmPFC$_{contrast}$ from *Figure 3d* as a seed and [Observer trial – No observer trial] as a psychological factor (see DM4). Specifically, individuals' BOLD responses were extracted from a 6 mm sphere centered on each individual's local maxima nearest to the group peak voxel of the dmPFC$_{contrast}$ cluster active at p<0.005 (x = –3, y=50, z=14; *Figure 3d*; *Supplementary file 1E*), and these time-series were then deconvolved and used as a physiological factor. Second, to examine whether the dmPFC and TPJ interacted during individuals' decision processes under social observation, we used the TPJ from *Figure 3c* as a seed and [Observer trial – No observer trial] as a psychological factor (see DM4). Given that the bilateral TPJ represented Prob(chosen) in the Observed phase, the left and the right TPJ were used for two separate PPI analyses, respectively (rTPJ: [x=63, y = –40, z=17], lTPJ: [x = –54, y = –37, z=14]; *Figure 3c*; *Supplementary file 1D*). The dmPFC$_{PPI}$ cluster from the result of the first PPI analysis (x=3, y=50, z=5, $k_E$ = 74; *Figure 4a*; *Supplementary file 1F*) was set as the target region. For the second-level regression, we added individuals' Social reliance parameter as a covariate.

## Mediation analyses

To examine whether the positive association between TPJ-dmPFC$_{PPI}$ connectivity and individuals' social reliance (log-transformed Social reliance estimates) (*Figure 4c*, *Figure 4—figure supplement 1a*) was mediated by the dmPFC's sensitivity to social cues (dmPFC$_{contrast}$-dmPFC$_{PPI}$ connectivity; *Figure 4a*), we conducted mediation analyses using an R package *mediation* (*Tingley et al., 2014*). Specifically, each individual's log-transformed Social reliance parameter was used as a predictor, dmPFC$_{contrast}$-dmPFC$_{PPI}$ connectivity was used as a mediator, and TPJ-dmPFC$_{PPI}$ connectivity was set as an outcome. Two separate mediation models were tested, with the outcome being the functional connectivity between the dmPFC and either the left or the right TPJ. The significance of the effects was assessed using a non-parametric bootstrapping method with 5000 samples, and an alpha level of 0.05 was applied to determine statistical significance.

## Acknowledgements

This work was supported in part by the National Research Foundation of Korea (NRF) (RS-2024–00420674 to DC) and the KBRI basic research program through Korea Brain Research Institute funded by Ministry of Science and ICT (24-BR-03–08 to DC).

## Additional information

### Funding

| Funder | Grant reference number | Author |
| --- | --- | --- |
| National Research Foundation of Korea | RS-2024-00420674 | Dongil Chung |
| Korea Brain Research Institute | 24-BR-03-08 | Dongil Chung |

The funders had no role in study design, data collection and interpretation, or the decision to submit the work for publication.

### Author contributions

HeeYoung Seon, Formal analysis, Investigation, Visualization, Writing – original draft, Writing – review and editing; Dongil Chung, Conceptualization, Supervision, Funding acquisition, Writing – original draft, Writing – review and editing

### Author ORCIDs

HeeYoung Seon ⓘ https://orcid.org/0000-0001-8620-7835

Dongil Chung [ID] https://orcid.org/0000-0003-1999-0326

### Ethics

Human subjects: All participants provided written informed consent. This study was approved by the Institutional Review Boards of the Ulsan National Institute of Science and Technology (UNISTIRB-19-45-A).

Reviewer #2 (Public review): https://doi.org/10.7554/eLife.102228.3.sa1
Reviewer #3 (Public review): https://doi.org/10.7554/eLife.102228.3.sa2
Author response https://doi.org/10.7554/eLife.102228.3.sa3

---

## Additional files

### Supplementary files

Supplementary file 1. Supplementary tables of behavioral and neuroimaging results.
MDAR checklist

### Data availability

Analytic scripts, behavioral data, and unthresholded first-level images are available on OSF.

The following dataset was generated:

| Author(s) | Year | Dataset title | Dataset URL | Database and Identifier |
| --- | --- | --- | --- | --- |
| Seon HY, Chung D | 2025 | Risky decision-making under social observation | https://doi.org/10. 17605/OSF.IO/3VBDA | Open Science Framework, 10.17605/OSF.IO/3VBDA |

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
