## [Editor Report · eLife Assessment]

Seon and Chung investigate changes in own risk-taking behavior, when they are being observed by a "risky" or "safe" player. Using computational modeling and model-informed fMRI, the authors present **convincing** evidence that participants adjust their choice congruent with the other player's type (either risky or safe). The conclusions of the paper are an **important** contribution to the field of social decision-making as they show a differentiated adjustment of choices and not just a universally riskier choice behavior when being observed as has been claimed in previous studies.

---

## [Referee Report · Reviewer #2 (Public review)]

Summary:

This study aims to investigate how social observation influences risky decision-making. Using a gambling task, the study explored how participants adjusted their risk-taking behavior when they believed their decisions were being observed by either a risk-averse or risk-seeking partner. The authors hypothesized that individuals would simulate the choices of their observers based on learned preferences and integrate these simulated choices into their own decision-making. In addition to behavioral experiments, the study employed computational modeling to formalize decision processes and fMRI to identify the neural underpinnings of risky decision-making under social observation.

Strengths:

The study provides a fresh perspective on social influence in decision-making, moving beyond the simple notion that social observation leads to uniformly riskier behavior. Instead, it shows that individuals adjust their choices depending on their beliefs about the observer's risk preferences, offering a more nuanced understanding of how social contexts shape decision-making. The authors provide evidence using comprehensive approaches, including behavioral data based on a well-designed task, computational modeling, and neuroimaging. The three models are well selected to compare at which level (e.g., computing utility, risk preference shift, and choice probability) the social influence alters one's risky decision-making. This approach allows for a more precise understanding of the cognitive processes underlying decision-making under social observation.

Weaknesses:

While the neuroimaging results are generally consistent with the behavioral and computational findings, the strength of the neural evidence could be improved. The authors' claims about the involvement of the TPJ and mPFC in integrating social information are plausible, but further analysis, such as model comparisons at the neuroimaging level, is needed to decisively rule out alternative interpretations that other computational models suggest.

My concern raised above in the previous round has been addressed with the newly added results. I now find the manuscript substantially improved.

I have only a minor suggestion: when discussing the conflict-related signals observed in the dACC and dlPFC, I encourage the authors to include alternative interpretations beyond conflict monitoring per se. For example, these signals may also reflect processes related to information updating during social learning or inference. While the study does not aim to dissociate these possibilities, acknowledging them would enrich the discussion and provide a broader perspective for readers.

Comments on revised version:

Thank you for the substantial revision. I believe the additional analyses have meaningfully strengthened the manuscript, particularly by improving the connection between the behavioral modeling and neuroimaging results. The findings are consistent with prior work while also providing novel insights.

When discussing the conflict-related signals observed in the dACC/dlPFC, I encourage the authors to include alternative interpretations in addition to conflict monitoring per se. For example, these signals may also reflect processes related to information updating during social learning or inference. While the study does not aim to dissociate these possibilities, acknowledging them would enrich the discussion and offer a broader perspective for readers.

I have updated my evaluation of the strength of evidence from Solid to Convincing.

---

## [Referee Report · Reviewer #3 (Public review)]

Summary:

This is an important paper using a novel paradigm to examine how observation affects social contagion of risk preferences. There is a lot of interest in the field on the mechanisms of social influence, and adding in the factor of whether observation also influences these contagion effects is intriguing.

Strengths:

There is an impressive combination of a multi-stage behavioural task as well as computational modelling and neuroimaging. The analyses are well conducted and the sample size is reasonable.

Comments on revised version:

Thank you for your helpful responses to my concerns. The manuscript is much improved and will make an important contribution to the literature. I have one remaining clarification. My request was for the authors to speculate in the discussion about lifespan differences in susceptibility to social influence, because the paper talks about how observing others' choices makes people riskier. I think it is important to explicitly acknowledge in the discussion that the sample tested was young adults, and it may be that the effects they observe are not the same in adolescents or older adults, as suggested in recent work (e.g. Reiter et al., 2019 Nat Comms, Su et al., 2024, Comms Psych). This is important to qualify general statements about how humans behave when observing others' risky decisions.

---

## [Author Response]

The following is the authors’ response to the original reviews.

**Reviewer #1 (Public review):**
Summary:Seon and Chung's study investigates the hypothesis that individuals take more risks when observed by others because they perceive others to be riskier than themselves. To test this, the authors designed an innovative experimental paradigm where participants were informed that their decisions would be observed by a "risky" player and a "safe" player. Participants underwent fMRI scanning during the task.Strengths:The research question is sound, and the experimental paradigm is well-suited to address the hypothesis.Weaknesses:I have several concerns. Most notably, the manuscript is difficult to read in parts, and I suggest a thorough revision of the writing for clarity, as some sections are nearly incomprehensible. Additionally, key statistical details are missing, and I have reservations about the choice of ROIs.

We appreciate the reviewer’s interest in and positive assessment of our work, and we thank the reviewer for the constructive feedback. In the current revision, we have revised the manuscript for clarity and added previously omitted statistical details. Furthermore, in the response letter, we have also provided additional explanations to clarify our approach, including the rationale for the choice and use of ROIs.

**Reviewer #2 (Public review):**
Summary:This study aims to investigate how social observation influences risky decision-making. Using a gambling task, the study explored how participants adjusted their risk-taking behavior when they believed their decisions were being observed by either a risk-averse or risk-seeking partner. The authors hypothesized that individuals would simulate the choices of their observers based on learned preferences and integrate these simulated choices into their own decision-making. In addition to behavioral experiments, the study employed computational modeling to formalize decision processes and fMRI to identify the neural underpinnings of risky decision-making under social observation.Strengths:The study provides a fresh perspective on social influence in decision-making, moving beyond the simple notion that social observation leads to uniformly riskier behavior. Instead, it shows that individuals adjust their choices depending on their beliefs about the observer's risk preferences, offering a more nuanced understanding of how social contexts shape decision-making. The authors provide evidence using comprehensive approaches, including behavioral data based on a well-designed task, computational modeling, and neuroimaging. The three models are well selected to compare at which level (e.g., computing utility, risk preference shift, and choice probability) the social influence alters one's risky decision-making. This approach allows for a more precise understanding of the cognitive processes underlying decision-making under social observation.Weaknesses:While the neuroimaging results are generally consistent with the behavioral and computational findings, the strength of the neural evidence could be improved. The authors' claims about the involvement of the TPJ and mPFC in integrating social information are plausible, but further analysis, such as model comparisons at the neuroimaging level, is needed to decisively rule out alternative interpretations that other computational models suggest.

We appreciate the reviewer’s interest in and positive assessment of our work, and we thank the reviewer for the constructive feedback. In the current revision, we have included neural results from additional analyses, which we believe provide stronger support for our proposed computational model.

**Reviewer #3 (Public review):**
Summary:This is an important paper using a novel paradigm to examine how observation affects the social contagion of risk preferences. There is a lot of interest in the field about the mechanisms of social influence, and adding in the factor of whether observation also influences these contagion effects is intriguing.Strengths:(1) There is an impressive combination of a multi-stage behavioural task with computational modelling and neuroimaging.(2) The analyses are well conducted and the sample size is reasonable.Weaknesses:(1) Anatomically it would be helpful to more explicitly distinguish between dmPFC and vmPFC. Particularly at the end of the introduction when mPFC and vmPFC are distinguished, as the vmPFC is in the mPFC.(2) The authors' definition of ROIs could be elaborated on further. They suggest that peaks are selected from neurosynth for different terms, but were there not multiple peaks identified within a functional or anatomical brain area? This section could be strengthened by confirming with anatomical ROIs where available, such as the atlases here http://www.rbmars.dds.nl/lab/CBPatlases.html and the Harvard-Oxford atlases.(3) How did the authors ensure there were enough trials to generate a reliable BOLD signal? The scanned part of the study seems relatively short.(4) It would be helpful to add whether any brain areas survived whole-brain correction.(5) There is a concern that mediation cannot be used to make causal inferences and much larger samples are needed to support claims of mediation. The authors should change the term mediation in order to not imply causality (they could talk about indirect effects instead) and highlight that the mediation analyses are exploratory as they would not be sufficiently powered (https://www.ncbi.nlm.nih.gov/pmc/articles/PMC2843527/).(6) The authors may want to speculate on lifespan differences in this susceptibility to risk preferences given recent evidence that older adults are relatively more susceptible to impulsive social influence (Zhu et al, 2024, comms psychology).

We appreciate the reviewer’s interest in and positive assessment of our work, and we thank the reviewer for the constructive feedback. In the response letter below, we address each of the reviewer’s comments, including clarifications regarding the ROIs and the limitations of the current study in interpreting the results.

**Reviewer #1 (Recommendations for the authors):**
(1) The neuroimaging hypotheses seem post hoc to me. First, the term "social inference" is used very loosely. In line 103 the authors mentioned that TPJ has been reported to be involved in inferring other's intentions and learning about others. However, in their task, it is not clear where inference is needed. All participants need to do is recall others' "preferences", rather than inferring a hidden variable or hidden intention. In addition, in some of the studies that the authors have cited (e.g., Park et al. 2021), the hippocampus is the focus of the inference, which gets no mention here.

How does solving this task require inference (as defined by the authors: inferring others' intentions)? And why do they choose TPJ while inference is not needed in this task?

We regret any confusion and would like to take this chance to clarify our hypothesis on social inference. As the reviewer pointed out, participants were indeed instructed to predict their choices, through which we expected them to learn the demonstrators’ preferences. Our computational model suggests that during the main phase of the task, i.e., the Observed phase, participants simulated others’ choices based on these previously learned risk preferences of others. The gamble choices they encountered (payoffs and associated probabilities) did not overlap with those in the Learning phase, and therefore, we expected that the cognitive process triggered by the social context involved active simulation—what we describe as making inference about others—rather than simple ‘recall’ of previously learned information. In line with this reasoning, we hypothesized that the TPJ, a brain region previously implicated in simulating others’ actions and intentions, would play a key role during the Observed phase.

Regarding the role of the hippocampus, the paper we cited by BoKyung Park et al. (2021), titled “The role of right temporoparietal junction in processing social prediction error across relationship contexts”, highlights the involvement of the rTPJ but does not mention the hippocampus. We are aware of the study by Seongmin A. Park et al. (2021), “Inferences on a multidimensional social hierarchy use a grid-like code”, which shows the involvement of the hippocampus and entorhinal cortex in making inferences about multidimensional social hierarchies; we believe the reviewer may have mistakenly assumed that we cited this article. As the study showed, the involvement of the hippocampus—and the use of its grid-like representation of social information—is likely tied to the multidimensional nature of task states. In our study, the hippocampus was not included as an ROI because we had no specific rationale to hypothesize that such grid-like representations would be recruited by our task.

(2) Social influence can be motivated informationally (to improve accuracy) or normatively (to be aligned with others). To me, it seems that the authors have studied the latter, because, first, there is no objectively correct response in this task and second, because participants changed their risk preference according to the preference of the observing partner. This distinction has not been made throughout the manuscript. This is important because the two process (information and normative) are supported by different neural processes and it is extremely useful to understand neural basis of which process the authors are studying.

We thank the reviewer for the opportunity to clarify the anticipated role of social influence in our study. As the reviewer pointed out, the gambling task used in our task does not have objectively correct or incorrect answers, and naturally, any social influence present during the task would align with normative social influence. To clarify this point, we have revised the discussion section as follows:

[Page 9, Line 345]

Observational learning and mimicry of others’ behavior are patterns commonly found in social animals, including nonhuman primates (Van de Waal et al., 2013). Such behaviors are thought to be driven either by a motivation to acquire additional information (‘informational conformity’) or by a motivation to align with group norm (‘normative conformity’), even when doing so does not necessarily lead to better outcomes (e.g., higher accuracy) (Cialdini & Goldstein, 2004). Given that there are no objectively correct or incorrect answers in the gambling task used in our study, the observed social influence is more consistent with normative conformity. However, we cannot rule out the possibility that individuals developed false beliefs about a particular observing partner—namely, that the partner had greater control over or insight into the gambling task. Future studies are needed to directly investigate whether individuals’ beliefs about others modulate informational social influence—that is, their motivation to use social information to gain additional insight by inferring others’ potential choices.

(3) From Line 160 onward, the authors report several findings without providing any effect sizes or statistics. Please add effect size and statistics for each finding.

We thank the reviewer for pointing this out. We have now added the corresponding effect sizes and statistical values for the reported findings, beginning from Line 160 in the revised manuscript.

(4) Line 270: "In particular, bilateral TPJ, brain regions not implicated in the Solo phase, positively tracked trial-by-trial model-estimated decision probabilities". How can the authors conclude that TPJ is not involved in the solo phase? As far as I understood from the text, TPJ was not included as one of the ROIs for analysis of the Solo phase. If it was included, it should be mentioned in the text and there should be a direct comparison between the effect sizes of the solo and the observer phase. If not, "not implicated in the Solo phase" is not justified and should be removed.

We apologize for the confusion. As the reviewer correctly pointed out, the TPJ was not included among the ROIs in our analysis of the Solo phase data; therefore, its involvement during the Solo phase was never directly assessed using an ROI-based approach.

To examine brain responses during the Observed phase, we first assessed whether regions that tracked decision probabilities during the Solo phase—vmPFC, vStr, and dACC—were also engaged in the Observed phase. The involvement of the TPJ during the Observed phase was revealed through a subsequent whole-brain analysis. To clarify this point, we now have revised the corresponding part as follows:

[Page 8, Line 276]

In particular, bilateral TPJ positively, brain regions not implicated in the Solo phase, tracked trial-by-trial model-estimated decision probabilities

à Notably, bilateral TPJ showed significant positive tracking of decision probabilities ~

(5) I am a bit puzzled about the PPI analysis. Is the main finding increased connectivity within mPFC in the observing condition? PPI is often done between two separate brain regions. I am not sure what it means that connectivity within mPFC increases in one condition compared to another. What was the motivation for this analysis? Can you also please explain what it means?

As the reviewer noted, psychophysiological interaction (PPI) analyses examine functional connectivity between brain regions as modulated by a psychological factor. To clarify our result, the reported ‘mPFC-mPFC connectivity’ refers to functional connectivity between the mPFC region responsive to the presence of an observing partner and an adjacent, anatomically distinct region within the mPFC. Note that we have revised the manuscript to refer to this region more specifically as the dorsomedial prefrontal cortex (dmPFC). Please see our response to Reviewer 3, Comment 1, for further details.

During the Observed phase of our task, social information was processed at two distinct time points. First, at the beginning of each decision trial, individuals were cued with the presence (or absence) of an observing partner (‘Partner presentation’). Second, the gamble options, as well as the observing partner’s identity, were revealed (‘Options revealed’). Because participants had previously learned about the observing partner’s risk preferences, we expected them to simulate the choice the partner would likely make. We hypothesized that if individuals indeed simulated the partner’s choice and incorporated this information into their decision-making process, the brain region involved in recognizing the partner’s presence (dmPFC_contrast_) would be functionally connected to the region responsible for integrating social information into the final decision (TPJ). Our results showed that the two regions were functionally connected via an indirect path through an anatomically adjacent cluster within the mPFC (dmPFC_PPI_). Given that the recognition of the partner’s presence and the simulation of their choice occurred at two distinct time points, we interpreted the functional connectivity between the two dmPFC clusters (dmPFC_contrast_ and dmPFC_PPI_) as evidence that the dmPFC_PPI_ remained engaged during the decision process to support simulation, rather than being involved solely in the passive recognition of the social context (i.e., observed vs not observed). Note that, consistent with this interpretation, functional connectivity was stronger in individuals who showed greater reliance on social information ('Social reliance' parameter in our model).

To avoid confusion, we have now labeled the two dmPFC clusters as dmPFC_contrast_—the seed region identified at partner presentation—and dmPFC_PPI_—the target region identified in the PPI analysis.

[Page 8, Line 284]

This cue was intended to dissociate neural responses to the social context per se (i.e., the presence of an observing partner), which we hypothesized would initiate social processing, from the neural processes involved in incorporating this information during the subsequent decision-making phase.

[Page 8, Line 291]

We tested whether the dmPFC was also involved in incorporating social information during the decision process under social observation, particularly among individuals who relied more heavily on simulating others’ behavior.

[Page 8, Line 297]

We confirmed that the functional connectivity between the dmPFC_contrast_ which is sensitive to cues regarding the presence of an observing partner, and its adjacent, anatomically distinct region within the dmPFC (‘dmPFC_PPI_’ hereafter; x = 3, y = 50, z = 5, k_E_ = .74, cluster-level P_FWE, SVC_ = 0.011; Fig. 4a, b, Table S5) was positively associated with individuals’ social reliance.

(6) In Line 107 the authors say "excitatory stimulation of the TPJ improved social cognition". Improved social cognition is too general and unspecific. Please be more specific.

We agree that the term ‘social cognition’ was too general and unspecific. In the revised manuscript, we have specified that the improvement was observed in tasks specifically involving the control of self-other representation, as demonstrated by Santiesteban et al. (2012).

[Page 4, Line 106]

Corroborating with these neuroimaging data, excitatory stimulation of the TPJ improved social cognition (Santiesteban et al., 2012),~

à Corroborating these neuroimaging findings, excitatory stimulation of the TPJ improved social cognition involving the control of self-other representation (Santiesteban et al., 2012),~

Writing:

We thank the reviewer for their thorough evaluation of our manuscript. We have now made the necessary revisions in accordance with the provided comments.

(7) Line 75: "one risky options" should be one risky option.

[Page 3, Line 74]

between one safe (i.e., guaranteed payoff) and one risky options.

between a safe option (i.e., guaranteed payoff) and a risky option.

(8) Line 82: were given with the same set of gamble should be "were given the same set of gamble".

[Page 3, Line 81]

In the third phase (‘Observed phase’), individuals were given with the same set of gamble choices they faced in the Solo phase,

In the third phase (‘Observed phase’), individuals were given the same set of gamble choices they faced in the Solo phase,~

(9) Line 63: and that the extent of such influence depends on the identity of the observer. It is not clear what the authors mean by the "identity of observer". Does it mean the preference of the observer?

Van Hoorn et al. (2018) showed that the degree of social influence varies depending on whether individuals are being observed by parents or by peers. While one might attribute this difference to divergent preferences typically held by parents and peers, it is important to note that other factors may also differ between these social groups. To avoid overinterpretation while preserving the original meaning, we have revised the sentence as follows:

[Page 3, Line 61]

However, recent studies showed that the unidirectional influence of social others’ presence may be also observed in adults (Otterbring, 2021), and that the extent of such influence depends on the identity of the observer (Van Hoorn et al., 2018).

However, recent studies showed that the unidirectional influence of social others’ presence can also be observed in adults (Otterbring, 2021), and that the extent of this influence depends on the observer’s identity—specifically, whether the observer is a parent or a peer (Van Hoorn et al., 2018).

(10) Line 103: "including inferring others' intention and in learning about others." An "in" is missing right before inferring.

[Page 4, Line 101]

The temporoparietal junction (TPJ) is another region known to play an important role in social cognitive functions, including inferring others’ intention and in learning about others (Behrens et al., 2008; Boorman et al., 2013; Charpentier et al., 2020; Park et al., 2021; Samson et al., 2004; Saxe & Kanwisher, 2003; Saxe & Kanwisher, 2013; Van Overwalle, 2009; Young et al., 2010).

The temporoparietal junction (TPJ) is another region known to play an important role in a range of social cognitive functions, including simulating others’ intention and choices, as well as learning about others (Behrens et al., 2008; Boorman et al., 2013; Charpentier et al., 2020; Park et al., 2021; Samson et al., 2004; Saxe & Kanwisher, 2003; Saxe & Kanwisher, 2013; Van Overwalle, 2009; Young et al., 2010).

(11) 106: "Corroborating with these neuroimaging data." It should be "corroborating these neuroimaging data".

[Page 4, Line 106]

Corroborating with these neuroimaging data, ~

Corroborating these neuroimaging findings, ~

(12) Lines 113-115. It is not clear what the authors are trying to say here.

We have now revised the sentence as follows:

[Page 4, Line 112]

We hypothesized that even if others’ choices are not explicitly presented, simple presence of social others may trigger inference about others’ potential choices, and the same set of brain regions will play an important role in value-based decision-making.

We hypothesized that, even in the absence of explicit information about others’ choices, the mere presence of social others could lead participants to conform to the option they believe others would choose. To do so, participants would need to simulate others’ potential choices, particularly when option values vary across trials. As a result, we propose that the same brain regions involved in simulating others’ decisions would also be engaged during value-based decision-making in the presence of social observers.

(13) Line 151: This sentence is too long and hard to follow:

We have now revised the sentence as follows:

[Page 5, Line 154]

Furthermore, individuals’ prediction responses on subsequent 10 prediction trials where no feedback was provided (Fig. 2b) as well as self-reports about the perceived riskiness of the partners collected at the end of the Learning phase (Fig. 1d) consistently showed that they were able to distinguish one partner from the other, and correctly estimate the partners’ risk preferences (Predicted risk preference: t(42) = -11.46, P = 1.66e-14; Self-report: t(42) = -35.83, P = 4.10e-33).

Furthermore, individuals’ prediction responses during the subsequent 10 trials without feedback consistently indicated that they could distinguish between the two partners and accurately estimate each partner’s risk preferences (t(42) = -11.46, P = 1.66e-14; Fig. 2b). Self-reported ratings of the partners’ perceived riskiness, collected after the Learning phase, further supported this finding (t(42) = -35.83, P = 4.10e-33; Fig. 1d).

(14) Line 178: This sentence is very hard to follow. I am not sure what the authors were trying to say here. Please clarify.

We have now revised the sentence as follows:

[Page 5, Line 183]

Various previous studies examined the impacts of social context on decision-making processes, but the suggested mechanisms by which individuals were affected by the social information depended on how the information was presented.

à Previous studies have shown that social context can influence decision-making processes. However, the underlying mechanisms proposed have varied depending on how the social information was presented.

(15) Line 183: "when individuals were given with the chances" should be "when individuals were given the chance".

[Page 5, Line 187]

On the contrary, when individuals were given with the chances~

On the contrary, when individuals were given the chances~

(16) Line 192: "are sensitive to the identity of the currently observing partner...". Do the authors mean are sensitive to the preferences of the currently observing partner? If so, please clarify, it is hard to follow.

We have now revised the sentence as follows:

[Page 5, Line 195]

We hypothesized that if individuals are sensitive to the identity of the currently observing partner, they would take into account the learned preferences of others in computing their choices rather than simply in guiding the direction how to change their own preferences.

à We hypothesized that if individuals are sensitive to the learned preferences of the observing partner, they would use this information to simulate the partner’s likely choices, rather than simply aligning their own preferences with those of the partner.

**Reviewer #2 (Recommendations for the authors):**
(1) The current neuroimaging findings appear to support the decision processes of all three models. I recommend that the authors provide more detailed evidence of model comparisons in the neuroimaging analysis. This should go beyond simply comparing the goodness of fit of neural activity.

We acknowledge that neuroimaging data alone often do not provide conclusive evidence for specific information processing. In our study, we examined brain regions that track decision probabilities and are associated with social cognition, such as simulating others’ choice tendencies. Because these processes are general and not tied to a specific computational model, neural responses supporting the occurrence of such processes cannot be used to rule out alternative decision models. For this reason, our approach prioritized a rigorous behavioral model comparison as a critical first step before probing the neural substrates underlying the proposed mechanism. Our behavioral model comparisons, including both quantitative fit indices and qualitative pattern predictions, indicated that the proposed model best accounted for participants' decision patterns across task conditions.

More importantly, to further validate the model, we conducted a model recovery analysis (see Fig. S2b in SI), which confirmed that our model can be reliably distinguished from alternative accounts even when behavioral differences are subtle. This result suggests that our model captures unique and meaningful characteristics of the decision process that are not equally well explained by competing models.

With this behavioral foundation, our neuroimaging analyses were designed not to serve as independent model arbiters, but rather to examine whether brain activity in regions of interest reflected the computations specified by the best-fitting model. We believe this two-step approach—first establishing behavioral validity, then linking model-derived variables to neural data—offers a principled framework for identifying the cognitive and neural mechanisms of decision-making.

Nevertheless, per the reviewer’s suggestion, we further examined whether there is neural encoding of both the participant’s own utility and the observer’s utility—serving as potential neural evidence to differentiate our model from the two alternative models. Please see below for our response to Reviewer 2’s Comment (2).

(2) Specifically, if participants are combining their own and simulated choices at the level of choice probability, we would expect to see neural encoding of both their own utility and the observer's utility. These may be observed in different areas of the mPFC, as demonstrated by Nicolle et al. (Neuron, 2012). In that study, decisions simulating others' choices were associated with activity in the dorsal mPFC, while one's own decisions were encoded in the vmPFC. On the contrary, if the brain encodes decision values based on the shifted risk preference, rather than encoding each decision's value in separate brain areas, this would support the alternative model.

We thank the reviewer for this constructive comment. In our Social reliance model, we assumed that the decision probability based on an individual’s own risk preferences, as well as that based on the observing partner’s risk preferences, both contribute to the individual’s final choice. As the reviewer suggested, neural evidence that differentiates our model from the two alternative models—the Risk preference change model and the Other-conferred utility model—would involve demonstrating neural encoding of both the participant’s own utility and the observer’s utility.

The utility differences between chosen and unchosen options from the two perspectives—self and observer—were highly correlated, preventing us from including both as regressors in the same design matrix. Instead, we defined ROIs along the ventral-to-dorsal axis of the mPFC, and examined whether each ROI more strongly reflected one’s own utility or that of the observer. Based on the meta-analysis by Clithero and Rangel (2014), we defined the most ventral mPFC ROI (ROI1) as a 10 mm-radius sphere centered at coordinate [x=-3, y=41, z=-7], a region previously associated with subjective value. From this ventral seed, we defined four additional spherical ROIs (10 mm radius each) at 12 mm intervals along the ventral-to-dorsal axis, resulting in five ROIs in total: ROI2 [x=-3, y=41, z=5], ROI3 [x=-3, y=41, z=17], ROI4 [x=-3, y=41, z=29], ROI5 [x=-3, y=41, z=41].

Consistent with Nicolle et al. (2012), the representation of one’s own utility (labelled as ‘Own subjective value’) and that of the observer (‘Observer’s subjective value’) was organized along the ventral-to-dorsal axis of the mPFC. Specifically, utility signals from the participant’s own perspective (SV_chosen, self_ – SV_unchosen, self_) were most prominently represented in the ventral-most ROIs (blue), whereas utility signals from the observer’s perspective (SV_chosen, observer_ – SV_unchosen, observer_) were most strongly represented in the dorsal-most ROIs (orange).

(3) Additionally, the authors may be able to detect neural signals related to conflict when the decisions of the individual and the observer differ, compared to when the decisions are congruent. These neural signatures would only be present if social influences are integrated at the choice level, as suggested by the authors.

If individuals simulate the choices that others might make, they may compare them with the choices they would have made themselves. To investigate this possibility, we categorized task trials as Conflict or No-conflict trials based on greedy choice predictions derived from a softmax decision rule. Conflict trials were those in which the choice predicted from the participant’s own risk preference differed from that predicted for the observer, whereas No-conflict trials involved the same predicted choice from both perspectives. A contrast between Conflict and No-conflict trials revealed that the dACC and dlPFC—regions previously associated with conflict monitoring and cognitive control (Shenhav et al., 2013)—were sensitive to differences in choice tendencies between the self and observer perspectives.

**Author response image 1. sa3fig1:** dACC and dlPFC are associated with the discrepancy between participants’ own choice tendencies and those of observing partners, as estimated based on prior beliefs about the partners’ risk preferences.

As the reviewer suggested, these results provide evidence in support of the Social Reliance model, which posits that participants simulate the observer's choice and integrate it with their own.

(4) Incorporating these additional analyses would provide stronger evidence for distinguishing between the models.

We again thank the reviewer for these constructive suggestions. Based on the new set of analyses and results, we have made the necessary revisions as noted above. We agree that these revisions provide stronger evidence for distinguishing between the models.

**Reviewer #3 (Recommendations for the authors):**
(1) Anatomically it would be helpful to more explicitly distinguish between dmPFC and vmPFC. Particularly at the end of the introduction when mPFC and vmPFC are distinguished, as the vmPFC is in the mPFC.

We appreciate the reviewer’s suggestion regarding the anatomical distinction between the dmPFC and vmPFC, particularly in relation to our use of the term “mPFC.” We acknowledge that the dmPFC and vmPFC are subregions of the broader mPFC. In our original manuscript, we referred to one region as mPFC in line with prior studies highlighting its role in social cognition and contextual processing (Behrens et al., 2008; Sul et al., 2015; Wittmann et al., 2016). However, in response to the reviewer’s comment and to more clearly distinguish this region from the ventral portion of the mPFC (i.e., vmPFC), which is canonically associated with subjective valuation, we have now revised the manuscript to refer to this region as the dmPFC. This terminology better reflects its association with social cognition, including model-estimated social reliance and sensitivity to social cues in our study.

(2) The authors' definition of ROIs could be elaborated on further. They suggest that peaks are selected from neurosynth for different terms, but were there not multiple peaks identified within a functional or anatomical brain area? This section could be strengthened by confirming with anatomical ROIs where available, such as the atlases here http://www.rbmars.dds.nl/lab/CBPatlases.html and the Harvard-Oxford atlases.

We appreciate the opportunity to clarify how our ROIs were defined. To identify the ROIs, we drew upon both prior literature and results from a term-based meta-analysis using Neurosynth. For each meta-map, we applied an FDR-corrected threshold of p < 0.01 and a cluster extent threshold of k ≥ 100 voxels to identify distinct functional clusters. For each cluster, we constructed a spherical ROI (radius = 10 mm) centered on its center of gravity. Note that for each anatomically distinct brain region, only a single center of gravity was identified and used to define the ROI. The resulting ROIs were subsequently used for small volume correction (SVC) in the second-level fMRI analyses.

For brain regions associated with decision-making processes, we obtained a meta-analytic activation map associated with the term “decision” from Neurosynth. After applying an FDR-corrected threshold of p < 0.001 and a cluster extent threshold of k ≥ 100 voxels, we identified five distinct clusters: vmPFC [x = -3, y = 38, z = -10]; right vStr [x = 12, y = 11, z = -7]; left vStr [x = -12, y = 8, z = -7]; dACC [x = 3, y = 26, z = 44]; and left Insula [x = -30, y = 23, z = -1]. To identify brain regions involved in decision-making under social observation, we used the Neurosynth meta-map associated with the term “social”, applying the same criteria (FDR p < 0.001, k ≥ 100). This analysis revealed several clusters, including bilateral TPJ: right TPJ [x = 51, y = -52, z = 14]; left TPJ [x = -51, y = -58, z = 17]. To isolate brain regions more specifically associated with social processing rather than valuation, we also constructed a conjunction map using the meta-maps for the terms “social” and “value.” We identified clusters present in the “social” map, but not in the “value” map. This analysis yielded, among others, a cluster in the dmPFC [x = 0, y = 50, z = 14].

To clarify our ROI analysis methods, we have now revised the manuscript to include more detailed information about the procedures used, as follows:

[Page 19, Line 746]

Region-of-interest (ROI) analyses. To define ROIs for the neural analyses conducted in the Observed phase, we used significant clusters identified during the Solo phase. Specifically, regions showing significant activation for Prob(chosen) in the DM0 (thresholded at P < 0.001) were selected as ROIs. Three ROI clusters were defined: the vStr (peak voxel at [x = 3, y = 14, z = -10], k_E_ = 9), vmPFC (peak voxel at [x = –3, y = 62, z = –13], k_E_ = 99), and dACC (peak voxel at [x = 12, y = 32, z = 29], k_E_ = 118). These ROIs were then applied in the Observed phase analyses to test whether similar neural representations are also engaged in social contexts.

Term-based meta-analytic maps from Neurosynth for small volume correction. To reduce the likelihood of false positives arising from random significant activations and to enhance sensitivity within regions of theoretical interest, small volume correction (SVC) was applied using term-based meta-analytic maps from Neurosynth. This approach allows for hypothesis-driven correction by restricting statistical testing to anatomically and functionally defined ROI. Specifically, three meta-analytic maps were generated using Neurosynth’s term-based analyses (Yarkoni et al., 2011), with a false discovery rate (FDR) corrected P < 0.01 and a cluster size > 100 voxels. For each resulting cluster, we defined a spherical ROI with a 10 mm radius centered on the cluster’s center of gravity. For each anatomically distinct brain region, only a single center of gravity was identified and used to define the corresponding ROI.

First, to identify regions encoding final decision probabilities during the Solo phase and enhance sensitivity, we used the meta-map associated with the term “decision” to identify neural substrates of value-based decision-making. This yielded three clusters: vmPFC ([x = -3, y = 38, z = -10]), vStr ([x = 12, y = 11, z = -7]), and dACC ([x = 3, y = 26, z = 44]) (Fig. 3a, S7). Second, to examine social processing during the Observed phase, we used the meta-map associated with the term “social” to identify brain regions typically involved in social cognition. This analysis revealed clusters, including the rTPJ ([x = 51, y = -52, z = 14]) and lTPJ ([x = -51, y = -58, z = 17]) (Fig. 3c, S8a). Third, to define an ROI involved in processing social cues independent of valuation, we used a meta-map associated with “social” but excluding “value”, isolating regions specific to non-valuation-related social cognition. This analysis revealed a cluster, including the dmPFC ([x = 0, y = 50, z = 14]) (Fig. 3d, 4a, S8b).

(3) How did the authors ensure there were enough trials to generate a reliable BOLD signal? The scanned part of the study seems relatively short.

We appreciate the reviewer’s concern regarding the number of trials and the potential implications for the reliability of the resulting BOLD signals. While we did not conduct formal statistical tests to determine the optimal number of trials, our task design, in general, followed well-established principles in functional neuroimaging. Specifically, we employed a jittered event-related design and used both temporal and dispersion derivatives in the GLM analyses. These strategies are widely recognized for enhancing the efficiency of BOLD signal deconvolution and improving model fit by accounting for inter-subject and inter-regional variability in the hemodynamic response function (HRF). Furthermore, the number of trials per condition in our study was comparable to those reported in previous publications (20-30 trials) that employed similar gambling paradigms to examine individual differences in the neural substrates of value-based decision-making (Chung et al., 2015; Chung et al., 2020).

(4) It would be helpful to add whether any brain areas survived whole-brain correction.

No brain regions survived whole-brain correction. Nevertheless, as described in the introduction, we had strong a priori hypotheses. Based on these hypotheses, we defined term-based ROIs using Neurosynth, and conducted small volume correction analyses. Per the reviewer’s suggestion, we have added information indicating that no brain regions survived whole-brain correction, as follows:

[Page 8, Line 281]

No additional regions survived whole-brain correction.

(5) There is a concern that mediation cannot be used to make causal inferences and much larger samples are needed to support claims of mediation. The authors should change the term mediation in order to not imply causality (they could talk about indirect effects instead) and highlight that the mediation analyses are exploratory as they would not be sufficiently powered (https://www.ncbi.nlm.nih.gov/pmc/articles/PMC2843527/).

We acknowledge the reviewer’s concerns regarding the causal interpretation of mediation analysis results. Per this comment, we have revised the manuscript as follows to avoid overinterpreting these results and to refrain from implying any causal inference.

[Page 9, Line 327]

Given that our sample size is smaller than the recommended threshold for detecting mediation effects (Fritz & MacKinnon, 2007), this significant indirect effect should be interpreted with caution, particularly with respect to causal inference.

(6) The authors may want to speculate on lifespan differences in this susceptibility to risk preferences given recent evidence that older adults are relatively more susceptible to impulsive social influence (Zhu et al, 2024, comms psychology).

We thank the reviewer for the thoughtful suggestion—we believe the referenced work is Zhilin Su et al. (2024). As noted in our manuscript, all participants in the current study were young adults aged between 18 and 29 years. Given this limited age range, our dataset does not provide sufficient variability to directly examine age-related differences across the lifespan. However, we are planning a follow-up study using the same task with older adult participants, which we believe will provide a valuable opportunity to address this important gap in understanding susceptibility to social influence across the lifespan.